# Pesticide use negatively affects bumble bees across European landscapes

Charlie C. Nicholson[1,18 ✉], Jessica Knapp[1,2,18 ✉], Tomasz Kiljanek[3], Matthias Albrecht[4], Marie-Pierre Chauzat[5], Cecilia Costa[6], Pilar De la Rúa[7], Alexandra-Maria Klein[8], Marika Mänd[9], Simon G. Potts[10], Oliver Schweiger[11,12], Irene Bottero[2], Elena Cini[10], Joachim R. de Miranda[13], Gennaro Di Prisco[6,14], Christophe Dominik[11,12], Simon Hodge[2], Vera Kaunath[1], Anina Knauer[4], Marion Laurent[15], Vicente Martínez-López[7], Piotr Medrzycki[6], Maria Helena Pereira-Peixoto[8], Risto Raimets[9], Janine M. Schwarz[4], Deepa Senapathi[10], Giovanni Tamburini[8,16], Mark J. F. Brown[17], Jane C. Stout[2] & Maj Rundlöf[1 ✉]

Sustainable agriculture requires balancing crop yields with the effects of pesticides on non-target organisms, such as bees and other crop pollinators. Field studies demonstrated that agricultural use of neonicotinoid insecticides can negatively affect wild bee species[1,2], leading to restrictions on these compounds[3]. However, besides neonicotinoids, field-based evidence of the effects of landscape pesticide exposure on wild bees is lacking. Bees encounter many pesticides in agricultural landscapes[4–9] and the effects of this landscape exposure on colony growth and development of any bee species remains unknown. Here we show that the many pesticides found in bumble bee-collected pollen are associated with reduced colony performance during crop bloom, especially in simplified landscapes with intensive agricultural practices. Our results from 316 *Bombus terrestris* colonies at 106 agricultural sites across eight European countries confirm that the regulatory system fails to sufficiently prevent pesticide-related impacts on non-target organisms, even for a eusocial pollinator species in which colony size may buffer against such impacts[10,11]. These findings support the need for postapproval monitoring of both pesticide exposure and effects to confirm that the regulatory process is sufficiently protective in limiting the collateral environmental damage of agricultural pesticide use.

Reliance on chemical pest control has created contaminated agricultural landscapes that expose bees to many pesticides[4–9,12]. Agricultural uses of neonicotinoid insecticides have been in the spotlight for their negative effects on bees[1,2,13,14] but it is unknown how effects scale beyond single substances in focal fields. We still do not know the consequences of landscape-level pesticide exposure, which results from agricultural uses of multiple approved pesticides over pollinator-relevant spatiotemporal scales, on the growth and development of any bee species. Here we empirically test the effects of landscape pesticide exposure on the key wild and commercial bumble bee pollinator *Bombus terrestris* L., answering recent calls for realistic pesticide mixture risk assessment at landscape scales[15].

As central place foragers, the fitness of bees depends on the net value of forage resources in their foraging range, which can be reduced if these resources are contaminated with hazardous pesticides[7,8,16]. Thus, intensively managed agricultural landscapes, with fewer flowers and seminatural habitats and simplified cropping systems with increased reliance on pesticides, are likely to increase the risk of pesticide exposure to bees[8,17,18]. Likewise, crops with different pesticide-use regimes and attractiveness to pollinators will also influence the exposure and risk of pesticides for bees[7,19]. To empirically test the consequences of landscape pesticide exposure, we placed sentinel colonies of *B. terrestris* (*n* = 316) along a gradient of the proportion cropland in the surrounding landscape (range 3–98%) at agricultural sites growing two focal flowering crops (apple *n* = 50 and oilseed rape *n* = 56) across eight European countries (Fig. 1a). We collected pollen samples from the colonies, which were screened for 267 compounds (Supplementary Table 1) to quantify pesticide residues.

We tracked bumble bee colony performance by weighing colonies before, during and after focal crop bloom and by counting all bees

[1]Department of Biology, Lund University, Lund, Sweden. [2]School of Natural Sciences, Trinity College Dublin, Dublin, Ireland. [3]Department of Pharmacology and Toxicology, National Veterinary Research Institute, Puławy, Poland. [4]Agroscope, Agroecology and Environment, Zurich, Switzerland. [5]Laboratory for Animal Health, ANSES, Paris-Est University, Maisons-Alfort, France. [6]Council for Agricultural Research and Economics—Agriculture and Environment Research Centre, Bologna, Italy. [7]Department of Zoology and Physical Anthropology, University of Murcia, Murcia, Spain. [8]Nature Conservation and Landscape Ecology, University of Freiburg, Freiburg, Germany. [9]Institute of Agricultural and Environmental Sciences, Estonian University of Life Sciences, Tartu, Estonia. [10]Centre for Agri-Environmental Research, School of Agriculture, Policy and Development, University of Reading, Reading, UK. [11]Department of Community Ecology, Helmholtz Centre for Environmental Research—UFZ, Halle, Germany. [12]German Centre for Integrative Biodiversity Research (iDiv) Halle-Jena-Leipzig, Leipzig, Germany. [13]Department of Ecology, Swedish University of Agricultural Sciences, Uppsala, Sweden. [14]Institute for Sustainable Plant Protection, The Italian National Research Council, Portici, Italy. [15]Unit of Honey Bee Pathology, Sophia Antipolis Laboratory, ANSES, Sophia Antipolis, France. [16]Department of Soil, Plant and Food Sciences, University of Bari, Bari, Italy. [17]Department of Biological Sciences, Royal Holloway University of London, Egham, UK. [18]These authors contributed equally: Charlie C. Nicholson, Jessica Knapp. ✉e-mail: charlie.nicholson@biol.lu.se; knappj@tcd.ie; maj.rundlof@biol.lu.se

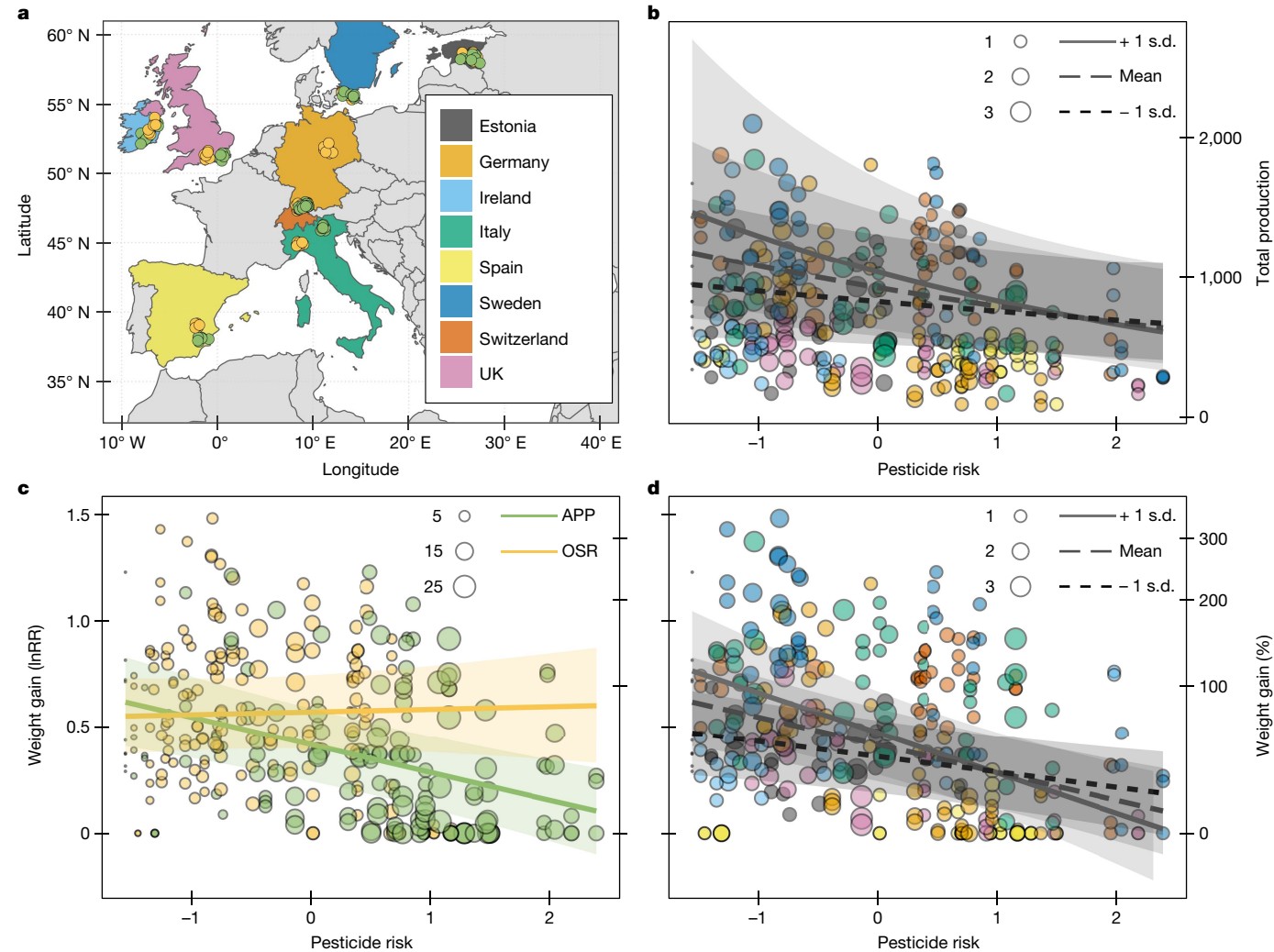

**Fig. 1 | Effects of landscape exposure to pesticides on bumble bee colony weight and production. a**, We deployed bumble bee (*B. terrestris* L.) colonies (*n* = 316) adjacent to apple (APP, green points) and oilseed rape (OSR, yellow points) across eight European countries. **b**, Colony production (total number of produced bees estimated by the sum of closed and eclosed cocoons) declined with pesticide risk (log-transformed and centred toxicity-weighted pesticide concentrations in pollen stores; Methods). **c,d**, Colony weight gain (response ratio ln($g_{max}/g_{initial}$) and percentage change (exp(lnRR)) also declined with pesticide risk (note double and shared *y* axes). Focal crop (**c**) and landscape

context (**b,d**) modified these effects, with stronger declines at apple (**c**; green line) compared to at oilseed rape sites (**c**; yellow line) and in landscapes with more cropland (**b,d**; solid line +1 s.d. proportion of cropland). Point colours (**b,d**) correspond to country colours (**a**) and are scaled by their MCR, the factor by which the mixture of compounds in a sample is riskier than the single most risky compound (Methods). Points in **c** are scaled by the number of pesticide compounds quantified in a sample. Fitted lines are estimated on the basis of generalized (**b**) and linear (**c,d**) mixed effects models. Shaded areas represent the regression 95% CI. Results from statistical models are given in Table 1.

at colony termination after bloom. We relate these colony performance endpoints to pesticide risk (summed toxicity-weighted pesticide concentrations in pollen; Methods) resulting from landscape exposure (Extended Data Fig. 1). We found that increasing pesticide risk reduced bumble bee colony production (summed eclosed and closed cocoons of all castes; Methods) and this effect was modified by an interaction with the proportion of cropland in the surrounding landscape (Fig. 1b and Table 1; generalized linear mixed effects model (GLMM): $\chi^2$ (1, 307) = 5.46, *P* = 0.019). Gain in colony weight—a metric inclusive of bees, brood and food—also decreased with increasing pesticide risk and focal crop (Fig. 1c; linear mixed effect model (LMM): $\chi^2$ (1, 307) = 9.13, *P* = 0.0025), as well as the proportion of cropland in the surrounding landscape (Fig. 1d; LMM: $\chi^2$ (1, 307) = 10.60, *P* = 0.001) modified this effect (Table 1). Colony weight gain was smaller with increasing pesticide risk when apple was the focal crop (slope estimate (95% confidence interval [CI]): −0.13 [−0.19, −0.07]) but not at the more resource-yielding oilseed rape[20] (0.02 [−0.06, 0.08]; Fig. 1c), suggesting that higher flower resource availability can mitigate the

negative effects of pesticides on bees[8,21]. Colony production (Fig. 1b) and weight gain (Fig. 1d) decreased more with increasing pesticide risk in landscapes with a higher proportion of cropland (more than 75%) compared to a lower proportion of cropland (less than 34%). Simplified landscapes, dominated by non-flowering cropland, generally contain fewer flower resources[22,23], potentially stressing colonies and interacting with pesticide effects[24,25]. Likewise, high pesticide risk may hamper the bees' foraging efficiency[26], an already difficult task in resource-poor environments.

Colony pollen stores contained many pesticides (95% with more than 1 compound; median 8; range 1–27), with more unique compounds in apple (80) than in oilseed rape (68). Although fungicides comprised 81% of total residues (μg kg$^{-1}$) and 62% of compound quantifications, insecticides represented most of the risk, with 99% of risk coming from nine insecticide compounds (Table 2). These high-risk compounds included the known bee health antagonists imidacloprid and indoxacarb, as well as pyrethroids and organophosphates (Table 2). Most pollen samples (62%) have maximum cumulative ratios (MCRs)—the factor by which

**Table 1 | Effects of pesticide risk, crop identity and proportion of cropland on colony production and weight gain**

| | Production of bees | | Weight gain | |
|---|---|---|---|---|
| | $\chi^2$ | P | $\chi^2$ | P |
| Initial weight | 0.22 | 0.63000 | 20.86 | <0.0001 |
| Risk | 4.99 | 0.02500 | 10.77 | 0.0010 |
| Crop | 12.26 | 0.00046 | 8.10 | 0.0044 |
| Cropland | 1.35 | 0.25000 | 1.31 | 0.2500 |
| Risk×crop | 3.98 | 0.04500 | 9.14 | 0.0025 |
| Risk×cropland | 5.46 | 0.01900 | 10.60 | 0.0011 |
| Crop×cropland | 4.95 | 0.02600 | 3.47 | 0.0620 |
| Risk×crop×cropland | 0.79 | 0.37000 | 0.60 | 0.4400 |
| $R^2$ marginal | 0.19 | | 0.23 | |
| $R^2$ conditional | 0.58 | | 0.76 | |

Colony total production of bees (sum of eclosed and intact cocoons) and weight gain (log response ratio) in relation to initial colony weight, pesticide risk (sum of toxicity-weighted pesticide concentrations; Methods), crop type (oilseed rape and apple), proportion of cropland in the surrounding landscape (1 km radius) and their interactions.

risk from all compounds was greater than its most risky compound (Methods)—less than 1.5 (range 1–3.8) (Fig. 1b,d). Together, these results indicate that pollen stores often contain many pesticides but that high concentrations of a few highly toxic insecticide compounds determine most of the mixture pesticide risk (Supplementary Table 2).

Focal crop pollen contributed a substantial but variable portion of the colony pollen stores (22 ± 22% at apple sites and 28 ± 28% at oilseed rape sites; mean ± s.d.; Extended Data Fig. 2a) and was not related to the proportion of cropland in the landscape ($\chi^2$ (1, 103) = 0.25, P = 0.62). Colony pollen stores at apple sites contained more pesticide compounds (Fig. 1c and Extended Data Fig. 2b). Apple and other fruit crops generally have higher pesticide use[27] and thus higher pesticide risk for bees, than do annual arable crops[5,7] or diversified farmland with permanent grasslands[28,29]. This reliance on many pesticides for pest management may increase the co-occurrence of compounds with known synergies, such as azole fungicides or cholinesterase-inhibiting insecticides[30]. Thus, our risk metric may underestimate or overestimate the potency of pesticide mixtures in agricultural landscapes because it assumes risk additivity of mixtures. Nonetheless, synergism among pesticides is relatively rare[30] and assuming concentration addition is considered a reasonable starting point in regulatory risk assessment of mixtures[31].

Mass-flowering crops such as oilseed rape can increase bumble bee colony growth when not accounting for pesticide exposure[32], especially when flowering coincides with peak worker numbers[33]. Therefore, we specifically timed colony placement to coincide with focal crop bloom, so that colony performance could be influenced by the net value of the focal crop: its nutritional benefit, minus pesticide cost. Bumble bee colony weight gain correlates with total production (Extended Data Fig. 3a), including queens (Extended Data Fig. 3b) and males[34,35], so our findings suggest the potential for adverse effects of pesticides on reproduction and subsequent population dynamics of bumble bees[36]. Indeed, we see that the production of new queens declined with increasing risk similarly to weight gain (Extended Data Table 1 and Extended Data Fig. 4). However, our approach meant colonies were at sites for different durations (apple 36.3 ± 11.4 days (mean ± s.d.); oilseed rape 43.0 ± 12.2 days; Extended Data Fig. 5) depending on region- and crop-specific bloom periods, precluding examination of full reproductive output and weight dynamics over the complete colony cycle, which follows an exponential growth and decline[37].

Understanding how and to what extent different cropping patterns and landscape contexts put key pollinator species at risk is essential for accurate and reliable pesticide risk assessment[15,38]. Our findings from 106 landscapes across Europe confirm that agricultural pesticide use results in exposure to many pesticides that reduce bumble bee colony performance during crop bloom, especially in simplified landscapes. Furthermore, our results can guide future postapproval monitoring efforts of non-target effects from landscape pesticide exposure[39]. *Bombus terrestris* is a valuable sentinel of the broader bee community for monitoring pesticide exposure[7] and effect[1] because its life history traits, such as colony size and foraging capacity, are intermediate to *Apis mellifera* and most solitary bee species. Nonetheless, *B. terrestris* forms colonies that may buffer the severity of pesticide effects[10,11]. Thus, the effects observed in our study may be more severe for the numerous solitary- and smaller-colony bee species[40,41].

Our results provide robust, European-wide evidence that landscape pesticide exposure negatively affects non-target organisms in agricultural landscapes. Using the average maximum weight of low-risk colonies (that is, the 25th percentile of risk) as a baseline, we found that 60% of remaining colonies exceed a current suggested specific protection goal (SPG) for bumble bees (10% colony weight reduction[42]; Extended Data Fig. 6a) and that these colonies were more at risk (Extended Data Fig. 6b). Further, compared to low-risk colonies, we observed a 34% reduction in maximum weight (estimated mean difference 393 g; Extended Data Fig. 6c), 52% reduction in total production (410 individuals; Extended Data Fig. 6d) and a 47% reduction in

**Table 2 | Ten compounds found in the colony pollen stores posing most risk to bumble bees in European agricultural landscapes**

| Pesticide (type) | Chemical group | LD$_{50}$ mean | LOQ | Concentration mean | Concentration median | Concentration 90th percentile | Frequency | Compound risk |
|---|---|---|---|---|---|---|---|---|
| Indoxacarb (I) | Oxadiazine | 0.1560 | 5.0 | 1,310 | 57 | 3,380 | 17 (16%) | 1,430 |
| Spinosad (I) | Spinosyn | 0.0303 | 5.0 | 658 | 658 | 1,170 | 2 (2%) | 434 |
| Chlorpyrifos-Ethyl (I) | Organophosphate | 0.1090 | 5.0 | 282 | 13.9 | 561 | 9 (8%) | 233 |
| Deltamethrin (I) | Pyrethroid | 0.0358 | 5.0 | 68.80 | 68.8 | 117 | 2 (2%) | 38.50 |
| Dimethoate (I) | Organophosphate | 0.1000 | 1.0 | 31 | 15.4 | 77.3 | 11 (10%) | 34.10 |
| Imidacloprid (I) | Neonicotinoid | 0.0424 | 1.0 | 9.490 | 8.1 | 17.5 | 9 (8%) | 20.20 |
| Cyfluthrin (I) | Pyrethroid | 0.0255 | 1.0 | 41.50 | 41.5 | 41.5 | 1 (1%) | 16.30 |
| Dithianon (F) | Quinone | 62.700[a] | 50.0 | 3,300 | 244 | 12,900 | 25 (24%) | 12.60 |
| Etofenprox (I) | Pyrethroid | 0.2020 | 5.0 | 61.90 | 47.5 | 91.9 | 3 (3%) | 9.19 |
| Chlorpyrifos-Methyl (I) | Organophosphate | 0.1620 | 5.0 | 36.90 | 16.6 | 80.9 | 4 (4%) | 9.08 |

Pesticide identity, type (I, insecticide; F, fungicide), chemical group, toxicity (average acute oral and contact LD$_{50}$ (dose required to cause 50% mortality in the test population) for *A. mellifera* adults, considering worst case from 24, 48 and 72 h values, µg per bee[46]), limit of quantification (LOQ) (µg kg$^{-1}$), concentrations (mean, median, 90th percentile; µg kg$^{-1}$), frequency of quantification (number of sites out of 106 sites with positive samples) and individual compound risk (Methods) of the ten riskiest pesticide compounds in colony pollen stores.
[a]LD$_{50}$ based on limit test.

queen production (21 individuals; Extended Data Fig. 6e) in the high-risk group (that is, the 90th percentile of risk). Thus, the European pesticide regulatory system for pesticides is not sufficiently protective given this SPG, indicating the need for postapproval monitoring of landscape exposure and its effects[15,24,39]. However, field-based assessments, as we present here, require high amounts of replication[43] and post hoc sensitivity analysis shows that more than 150 colony–site combinations are required to detect the effects we observed (Extended Data Fig. 7). In silico approaches to predict bee health are promising for a more holistic environmental risk assessment[15], for which these results could form an empirical basis.

Our results show that ambitious sustainability goals related to pesticide reduction—objectives of the COP 15 meeting on the Convention on Biological Diversity[44] and the European Farm to Fork strategy[45]—would benefit bee populations and potentially the pollination services they provide[46]. Conversely, the current assumption of pesticide regulation—that chemicals that individually pass laboratory tests and semifield trials are considered environmentally benign—fails to safeguard bees and other pollinators that support agricultural production and wild plant pollination. Thus, future monitoring of bee populations under typical agricultural practices, accounting for landscape exposure, is a vital step towards a system of pollinator pesticidovigilance[39].

## Online content

1. Rundlöf, M. et al. Seed coating with a neonicotinoid insecticide negatively affects wild bees. *Nature* **521**, 77–80 (2015).
2. Woodcock, B. et al. Country-specific effects of neonicotinoid pesticides on honey bees and wild bees. *Science* **356**, 1393–1395 (2017).
3. Domenica, A. et al. Neonicotinoids and bees: the case of the European regulatory risk assessment. *Sci. Total Environ.* **579**, 966–971 (2017).
4. David, A. et al. Widespread contamination of wildflower and bee-collected pollen with complex mixtures of neonicotinoids and fungicides commonly applied to crops. *Environ. Int.* **88**, 169–178 (2016).
5. Graham, K. K. et al. Identities, concentrations and sources of pesticide exposure in pollen collected by managed bees during blueberry pollination. *Sci. Rep.* **11**, 16857 (2021).
6. Hladik, M. L., Vandever, M. & Smalling, K. L. Exposure of native bees foraging in an agricultural landscape to current-use pesticides. *Sci. Total Environ.* **542**, 469–477 (2016).
7. Knapp, et al. Ecological traits interact with landscape context to determine bees' pesticide risk. *Nat. Ecol. Evol.* **7**, 547–556 (2023).
8. Rundlöf, M. et al. Flower plantings support wild bee reproduction and may also mitigate pesticide exposure effects. *J. Appl. Ecol.* **59**, 2117–2127 (2022).
9. Ward, L. T. et al. Pesticide exposure of wild bees and honey bees foraging from field border flowers in intensively managed agriculture areas. *Sci. Total Environ.* **831**, 154697 (2022).
10. Crall, J. D., de Bivort, B. L., Dey, B. & Versypt, A. N. F. Social buffering of pesticides in bumblebees: agent-based modeling of the effects of colony size and neonicotinoid exposure on behavior within nests. *Front. Ecol. Evol.* **7**, 51 (2019).
11. Zaragoza-Trello, C., Vilà, M., Botías, C. & Bartomeus, I. Interactions among global change pressures act in a non-additive way on bumblebee individuals and colonies. *Funct. Ecol.* **35**, 420–434 (2021).
12. Sponsler, D. B. et al. Pesticides and pollinators: a socioecological synthesis. *Sci. Total Environ.* **662**, 1012–1027 (2019).
13. Cressey, D. The bitter battle over the world's most popular insecticides. *Nature* **551**, 156–158 (2017).
14. Woodcock, B. et al. Impacts of neonicotinoid use on long-term population changes in wild bees in England. *Nat. Commun.* **7**, 12459 (2016).
15. Topping, C. J., Aldrich, A. & Berny, P. Overhaul environmental risk assessment for pesticides. *Science* **367**, 360–363 (2020).
16. Rundlöf, M. & Lundin, O. Can costs of pesticide exposure for bumblebees be balanced by benefits from a mass-flowering crop? *Environ. Sci. Technol.* **53**, 14144–14151 (2019).
17. Stuligross, C. & Williams, N. M. Pesticide and resource stressors additively impair wild bee reproduction: stressors additively impair wild bees. *Proc. R. Soc. B Biol. Sci.* **287**, 20201390 (2020).
18. Tosi, S., Nieh, J. C., Sgolastra, F., Cabbri, R. & Medrzycki, P. Neonicotinoid pesticides and nutritional stress synergistically reduce survival in honey bees. *Proc. R. Soc. B Biol. Sci.* **284**, 20171711 (2017).
19. Graham, K. K. et al. Pesticide risk to managed bees during blueberry pollination is primarily driven by off-farm exposures. *Sci. Rep.* **12**, 7189 (2022).
20. Baude, M. et al. Historical nectar assessment reveals the fall and rise of floral resources in Britain. *Nature* **530**, 85–88 (2016).
21. Stuligross, C. & Williams, N. Past insecticide exposure reduces bee reproduction and population growth rate. *Proc. Natl Acad. Sci. USA* **118**, e2109909118 (2021).
22. Dicks, L. V. et al. A global-scale expert assessment of drivers and risks associated with pollinator decline. *Nat. Ecol. Evol.* **5**, 1453–1461 (2021).
23. Persson, A. S. & Smith, H. G. Seasonal persistence of bumblebee populations is affected by landscape context. *Agric. Ecosyst. Environ.* **165**, 201–209 (2013).
24. Siviter, H. et al. Agrochemicals interact synergistically to increase bee mortality. *Nature* **596**, 389–392 (2021).
25. Knauer, A. C. et al. Nutritional stress exacerbates impact of a novel insecticide on solitary bees' behaviour, reproduction and survival. *Proc. R. Soc. B Biol. Sci.* **289**, 20221013 (2022).
26. Gill, R. J. & Raine, N. E. Chronic impairment of bumblebee natural foraging behaviour induced by sublethal pesticide exposure. *Funct. Ecol.* **28**, 1459–1471 (2014).
27. Garthwaite, D. et al. Collection of pesticide application data in view of performing Environmental Risk Assessments for pesticides. *EFSA Support. Publ.* **12**, 846E (2017).
28. Böhme, F., Bischoff, G., Zebitz, C. P. W., Rosenkranz, P. & Wallner, K. Pesticide residue survey of pollen loads collected by honeybees (*Apis mellifera*) in daily intervals at three agricultural sites in South Germany. *PLoS ONE* **13**, e0199995 (2018).
29. Rondeau, S., Baert, N., McArt, S. & Raine, N. E. Quantifying exposure of bumblebee (*Bombus* spp.) queens to pesticide residues when hibernating in agricultural soils. *Environ. Pollut.* **309**, 119722 (2022).
30. Cedergreen, N. Quantifying synergy: a systematic review of mixture toxicity studies within environmental toxicology. *PLoS ONE* **9**, e96580 (2014).
31. EFSA Scientific Committee Guidance on harmonised methodologies for human health, animal health and ecological risk assessment of combined exposure to multiple chemicals. *EFSA J.* **17**, e05634 (2019).
32. Westphal, C., Steffan-Dewenter, I. & Tscharntke, T. Mass flowering crops enhance pollinator densities at a landscape scale. *Ecol. Lett.* **6**, 961–965 (2003).
33. Hovestadt, T., Mitesser, O., Poethke, A. & Holzschuh, A. Explaining the variability in the response of annual eusocial insects to mass-flowering events. *J. Anim. Ecol.* **88**, 178–188 (2019).
34. Lefebvre, D. & Pierre, J. Hive weight as an indicator of bumblebee colony growth. *J. Apic. Res.* **47**, 217–218 (2006).
35. Westphal, C., Steffan-Dewenter, I. & Tscharntke, T. Mass flowering oilseed rape improves early colony growth but not sexual reproduction of bumblebees. *J. Appl. Ecol.* **46**, 187–193 (2009).
36. Baron, G. L., Jansen, V. A. A., Brown, M. J. F. & Raine, N. E. Pesticide reduces bumblebee colony initiation and increases probability of population extinction. *Nat. Ecol. Evol.* **1**, 1308–1316 (2017).
37. Duchateau, M. J. & Velthuis, H. Development and reproductive strategies in *Bombus terrestris* colonies. *Behaviour* **107**, 186–207 (1988).
38. More, S. J., Auteri, D., Rortais, A. & Pagani, S. EFSA is working to protect bees and shape the future of environmental risk assessment. *EFSA J.* **19**, e190101 (2021).
39. Milner, A. M. & Boyd, I. L. Toward pesticidovigilance. *Science* **357**, 1232–1234 (2017).
40. Nieto, A. et al. *European Red List of Bees* (European Commission, 2014).
41. Wood, T. J. et al. Managed honey bees as a radar for wild bee decline? *Apidologie* **51**, 1100–1116 (2020).
42. EFSA. et al. Analysis of the evidence to support the definition of Specific Protection Goals for bumble bees and solitary bees. *EFSA Support. Publ.* **19**, 7125E (2022).
43. Woodcock, B. et al. Replication, effect sizes and identifying the biological impacts of pesticides on bees under field conditions. *J. Appl. Ecol.* **53**, 1358–1362 (2016).
44. *Updated Synthesis of the Proposals of Parties and Observers on the Structure of the Post-2020 Global Biodiversity Framework and its Targets,* www.cbd.int/conferences/post2020/submissions/2019-075 (CBD Secretariat, 2020).
45. *The European Green Deal* (The European Commission, 2019).
46. Stanley, D. A. et al. Neonicotinoid pesticide exposure impairs crop pollination services provided by bumblebees. *Nature* **528**, 548–550 (2015).

## Methods

### Study landscapes

Our site network spanned 128 agricultural sites in eight European countries encompassing many biogeographic zones with differing climates and seasonality[47] (Fig. 1a). Sites focused on either oilseed rape or apple crops. In each focal crop, sites were selected to occur along a gradient of proportion of cropland within 1 km radius landscapes. This proportion is an established proxy for the agricultural management intensity typical of each country[47]. We chose oilseed rape and apple as our focal crops to reflect annual and perennial cropping practices and, therefore, different pest pressures, pest management strategies and pesticide use[27,48]. Furthermore, these crops are grown throughout Europe and so provided standardization across this geographic range. Apple and oilseed rape provide abundant food resources for pollinators[49], require pollination[50] and are economically important[51], reiterating the need for reliable ecosystem services in these landscapes. The most dominant land cover types were cropland (mean 55%; range 3–98%) and semi-natural areas (mean 37%; range 0.1–93%), where the latter comprised grasslands (mean 19%; range 0.1–76%), woodlands (mean 18%; range 0–62%) and wetlands (mean 0.1%; range 0–3%). These two dominant land covers were strongly negatively correlated ($R_{104} = -0.95, P < 0.001$). All sites were more than 3 km apart to ensure the spatial independence of the bumble bee colonies, whose foraging range is generally less than 1.5 km (ref. 52).

### Sentinel colonies and measurements of colony performance during crop bloom

At each site, we used three bumble bee colonies (*B. terrestris terrestris* for continental Europe and *B. terrestris audax* for the UK and Ireland) ($n = 384$), housed in protective structures (Extended Data Fig. 8), before focal crop bloom in 2019. Before deployment, we confirmed that each colony had a natal queen and recorded its initial weight ($648 \pm 70.9$ g (mean ± s.d.); Extended Data Fig. 5c). Colonies were weighed again during peak bloom of the focal crop in each country. At the end of the crop bloom, colonies were weighed again, then sealed, retrieved from sites and frozen. Of the 384 colonies initially deployed across 128 sites (64 apple, 64 oilseed rape), we analysed 316 colonies from 106 sites. This reduced sample size is due to colony losses (for example, animal attack and overrun by machinery; $n = 5$) or colonies not yielding enough stored pollen material for pesticide quantification ($n = 63$; Supplementary Table 3). The last could potentially be avoided in any future studies by complementing with concurrent collection of returning foragers' corbicular pollen[7,8].

In the laboratory, we removed any wax covering and sorted through the colony structure to count the number of intact and eclosed worker/male and queen cocoons (Extended Data Fig. 8), on the basis of their different size[1]. Our approach allowed us to derive two main indices of colony performance: (1) colony weight gain and (2) the total colony production. For weight gain, we calculated the natural-log response ratio for each colony as $\ln(g_{max}/g_{initial})$, where $g_{initial}$ is weight before bloom and $g_{max}$ is the maximum weight achieved by a colony during its field placement. In most cases (62% of colonies), $g_{max}$ was achieved by the final weighing but, in some cases, $g_{max}$ was achieved at the second (26%) or first (12%) weighing. For total colony production, we summed the number of intact and eclosed cocoons, including the eclosed cocoons used for nectar and pollen storage, instead of the number of bee individuals present at the time of colony termination, as new reproductives (gynes and males) could have left the colony at the time of retrieval. In addition, we summed the number of intact and eclosed queen cocoons for an indication of the colony reproduction. Colony termination was timed to crop bloom, rather than colony dynamics, preventing colony cycle completion and full reproduction. Queen production should therefore be interpreted with caution. During colony dissection, we extracted pollen stored in colonies (Extended Data Fig. 8),

pooling from all three colonies aiming for at least 15 g but using samples down to 0.52 g for pesticide residue analysis ($n = 106$ pollen samples). Samples were homogenized before preparing subsamples for palynological and pesticide residue analyses. All samples were stored at −20 °C.

### Palynological analysis

Palynological analyses were performed at the Research Centre for Agriculture and Environment (CREA) Bologna, Italy. For each homogenized pollen store sample, 1.0 g was dissolved in 20 ml of distilled water. Using a Pasteur pipette, a drop of sediment was placed on a microscope slide and spread out over an area about $18 \times 18$ mm². After drying, the sediment was included in glycerine jelly and covered with the cover slip. Examination under the microscope was performed with ×400 magnification. After a first read to identify all the pollen types in the slide, a second read of the slide was carried out until 500 pollen grains were counted. Abortive, irregular or broken pollen grains were counted if they could be identified. Non-identifiable or non-identified grains were noted separately. Recognition of pollen type was based on comparison between the observed pollen forms and those present in the CREA collection of reference slides (a database with more than 1,000 thermophilous species developed using anthers of identified plant species). For each pollen type, the percentage with respect to the total number of counted pollen grains was calculated.

### Pesticide residue analysis

Pesticide residue analyses were performed at the Department of Pharmacology and Toxicology, National Veterinary Research Institute, Puławy, Poland, which is the National Reference Laboratory for pesticide residue analysis and regularly participates in international proficiency tests with satisfactory results. We used 0.3 g of homogenized pollen store samples to screen for 267 compounds including isomers and metabolites (Supplementary Table 1). Particular attention was paid to analysing pesticides that are the active substances in plant protection products recommended for the protection of oilseed rape and apple orchards[53,54]. We use a previously described method[55] that is validated according to SANTE/12682/2019 (ref. 56) and accredited in accordance with the ISO 17025 standard. First, a sample was extracted with 1 ml of a solution of 5% formic acid in acetonitrile, and then the ammonium formate salt was added. The extract was subjected to clean-up by freezing and two-step dispersive solid phase extraction with a Supel QuE Verde sorbents. After first step dispersive solid phase extraction (dSPE), a portion of extract was analysed by liquid chromatography tandem mass spectrometry system (Agilent 1260 HPLC coupled with an AB Sciex QTRAP 6500 mass spectrometer) for 200 pesticide residues. The remaining extract was subjected to second step dSPE clean-up by another Supel QuE Verde and then, after concentration and solvent exchange, was analysed by gas chromatography tandem mass spectrometry (Agilent GC 7890 A+ coupled with a 7000B mass spectrometer) for another 61 pesticide and 6 ndl-PCB residues. Procedural standard calibration was used for calibration[56]. Reagent blanks and blank samples were analysed in each batch. Recovery checks with samples spiked with pesticides at limit of quantification (LOQ) levels were performed in each analytical batch to meet SANTE/12682/2019 criteria[56].

### Calculation of pesticide risk

We use toxicity-weighted concentrations (TWC) as a basis for indicating the direct pesticide risk to bees[7,8], where the TWC for each compound ($TWC_i$) is the ratio of its concentration in bee-collected pollen ($\mu g\ kg^{-1}$; $c_i$) and its respective acute toxicity endpoint ($LD_{50i}$—the dose required to cause 50% mortality in the test population). Following a concentration addition approach, the recommended default for

mixture environmental risk assessment[31,57], we summed TWCs to calculate the additive toxicity-weighted concentration of all compounds within a sample per site ($TWC_{mix}$):

$$TWC_{mix} = \sum_{i=1}^{n} \frac{c_i}{LD_{50}i}$$

We used an average of the acute oral and contact lethal doses $LD_{50}$ for each compound sourced from the Pesticide Properties Database[58,59] to provide an overall indicator of toxicity, reflective of how bees encounter pesticides in the landscape, that is, moving contaminated food in contact with their bodies for oral consumption[60]. We used the $LD_{50}$ for adult *A. mellifera* because there are incomplete toxicity data for other bee species and, if there are data, intertaxa correlation is high[60,61]. We rounded $LD_{50}$ down when based on limit tests and expressed as 'greater than'[58]. All values less than LOQ are treated as zero.

We quantify individual compound risk (Table 2 and Supplementary Table 2) as the average of concentrations for a given compound divided by its respective $LD_{50}$ and multiplied by its site detection frequency[62]. To calculate the dominance of individual compounds to the mixture risk, we determine the MCR of each pollen sample as the additive toxicity-weighted concentration of the mixture ($TWC_{mix}$) divided by the highest toxicity-weighted concentration of a single mixture component ($\max(TWC_i)$)[63]

$$MCR = \frac{TWC_{mix}}{\max(TWC_i)}$$

When MCR = 1, risk comes from a single compound; thus, the MCR represents the factor by which the pesticide mixture is riskier than the single most risky compound.

## Statistical analyses

We tested whether pesticide risk ($TWC_{mix}$) interacts with crop type and proportion cropland to affect our measures of colony performance (total colony production, weight gain, maximum weight and queen production). Given a strong right skew, we log-transformed ($\ln(x + 0.1)$) risk values. We centred risk and cropland values to aid the interpretation of interaction terms. For weight gain, we specified an LMM with risk, crop type, proportion cropland and their interactions as fixed effects and with site nested in country as a random effect. We specified a GLMM with a negative binomial error distribution for overdispersed count data of total colony production (dispersion ratio = 54.98; $P < 0.001$). We used the same fixed and random effect structure as above. We analysed two more measures of colony performance: maximum weight and queen cocoon production (total of intact and eclosed queen cocoons). We specified an LMM as above with weight log-transformed because it improved diagnostics of model residuals and our results are qualitatively similar if weight is untransformed (Extended Data Table 1). We specified a GLMM as above for queen cocoon production and with a single, constant zero-inflation parameter (Extended Data Table 1). We included initial colony weight ($g_{initial}$) as a covariate in the above models to account for variation in colony starting conditions. Models showed little multicollinearity (VIF range 1.03–3.28 across all models) and we confirmed that risk and proportion of cropland were independent by means of an LMM with the country as a random effect (marginal $R^2$ ($R^2$m) = 0.02; $\chi^2 = 1.38$, $P = 0.24$; Extended Data Fig. 9a). We also confirmed that risk was independent of initial colony weight by means of an LMM with the country as a random effect ($R^2$m = 0.01; $\chi^2 = 1.22$, $P = 0.27$; Extended Data Fig. 9b). We performed analyses and data visualization using R v.4.1.1. We constructed LMMs with the lme4 package[64] and GLMMs with the glmmTMB package[65]. We report $R^2$ values calculated following the methods of ref. 66. We estimated marginal means with the emmeans package[67].

We estimated interaction slopes with the interactions package[68] and evaluated models for overdispersion, normality and multicollinearity using diagnostic functions in the performance package[69] and the DHARMa package[70].

## Reporting summary

Further information on research design is available in the Nature Portfolio Reporting Summary linked to this article.

## Data availability

The datasets analysed for the current study are available through the PoshBee project (Deliverable D1.6 Database of field records), the Pesticide Properties Database and through figshare: https://doi.org/10.6084/m9.figshare.24235573.

47. Hodge, S. et al. Design and planning of a transdisciplinary investigation into farmland pollinators: rationale, co-design and lessons learned. *Sustainability* **14**, 10549 (2022).
48. Nicholson, C. C. & Williams, N. M. Cropland heterogeneity drives frequency and intensity of pesticide use. *Environ. Res.* **16**, 074008 (2021).
49. Holzschuh, A., Dormann, C. F., Tscharntke, T. & Steffan-Dewenter, I. Mass-flowering crops enhance wild bee abundance. *Oecologia* **172**, 477–484 (2013).
50. Klein, A.-M. et al. Importance of pollinators in changing landscapes for world crops. *Proc. R. Soc. B Biol. Sci.* **274**, 303–313 (2007).
51. Leonhardt, S. D., Gallai, N., Garibaldi, L. A., Kuhlmann, M. & Klein, A. M. Economic gain, stability of pollination and bee diversity decrease from southern to northern Europe. *Basic Appl. Ecol.* **14**, 461–471 (2013).
52. Kendall, L. K. et al. The potential and realized foraging movements of bees are differentially determined by body size and sociality. *Ecology* **103**, e3809 (2022).
53. Bryk, H. et al. *Apple Trees Protection Program* (Research Institute of Horticulture, 2018).
54. Kierzek, R. et al. *Winter Oilseed Rape Protection Program* (Institute of Plant Protection–National Research Institute, 2018).
55. Kiljanek, T., Niewiadowska, A., Małysiak, M. & Posyniak, A. Miniaturized multiresidue method for determination of 267 pesticides, their metabolites and polychlorinated biphenyls in low mass beebread samples by liquid and gas chromatography coupled with tandem mass spectrometry. *Talanta* **235**, 122721 (2021).
56. *Analytical Quality Control and Method Validation Procedures for Pesticide Residues Analysis in Food and Feed*. SANTE/12682/2019 (European Commission, 2020).
57. Bopp, S. K. et al. Current EU research activities on combined exposure to multiple chemicals. *Environ. Int.* **120**, 544–562 (2018).
58. *Pesticide Properties Data Base* (University of Hertfordshire, 2022).
59. Lewis, K. A., Tzilivakis, J., Warner, D. J. & Green, A. An international database for pesticide risk assessments and management. *Hum. Ecol. Risk Assess.* **22**, 1050–1064 (2016).
60. Arena, M. & Sgolastra, F. A meta-analysis comparing the sensitivity of bees to pesticides. *Ecotoxicology* **23**, 324–334 (2014).
61. DiBartolomeis, M., Kegley, S., Mineau, P., Radford, R. & Klein, K. An assessment of acute insecticide toxicity loading (AITL) of chemical pesticides used on agricultural land in the United States. *PLoS ONE* **14**, e0220029 (2019).
62. Sanchez-Bayo, F. & Goka, K. Pesticide residues and bees—a risk assessment. *PLoS ONE* **9**, e94482 (2014).
63. Price, P. S. & Han, X. Maximum cumulative ratio (MCR) as a tool for assessing the value of performing a cumulative risk assessment. *Int. J. Environ. Res. Public Health* **8**, 2212–2225 (2011).
64. Bates, D., Mächler, M. & Bolker B, W. S. Fitting linear mixed-effects models using lme4. *J. Stat. Softw.* **67**, 1–48 (2015).
65. Brooks, M. E. et al. glmmTMB balances speed and flexibility among packages for zero-inflated generalized linear mixed modeling. *R J.* **9**, 378–400 (2017).
66. Nakagawa, S. & Schielzeth, H. A general and simple method for obtaining $R^2$ from generalized linear mixed-effects models. *Methods Ecol. Evol.* **4**, 133–142 (2013).
67. Lenth, R. V. emmeans: Estimated Marginal Means, aka Least-Squares Means. R package version 1.7.2. https://CRAN.R-project.org/package=emmeans (2022).
68. Long J. A. interactions: Comprehensive, user-friendly toolkit for probing interactions. R package version 1.1.6 (2022).
69. Lüdecke, D., Ben-shachar, M. S., Patil, I. & Makowski, D. performance: an R package for assessment, comparison and testing of statistical models statement of need. *J. Open Source Softw.* **6**, 3139 (2021).
70. Hartig, F. DHARMa: residual diagnostics for hierarchical (multi-level/mixed) regression models. *GitHub* http://florianhartig.github.io/DHARMa/ (2021).

**Acknowledgements** We thank farmers and landowners for access to their land. We also thank A. Bates, J. Borth, M. Dietenberger, M. Cotter, R. George, L. Junk, S. Kivelitz, S. Lotz, J. Panziera, B. Rai, B. Schaer, G. Svensson and A. Turner for field and laboratory assistance; O. Burek, M. Goliszek, P. Łusiak, M. Małysiak, A. Niewiadowska and S. Semeniuk for laboratory assistance in pesticide residue analysis; M. T. N. N. Huyen for pesticide data curation; A. Dalpiaz, K. Ivarsson, L. Marchel and A. Saccardo for site selection and design; A. Neubauer for graphic design; and G. Turney for project management. This project received funding from the European Horizon 2020 research and innovation programme under grant agreement no. 773921. C.C.N., J.K. and

M.R. were supported by the Swedish research council Formas (grant 2018-02283) and the Strategic Research Area 'Biodiversity and Ecosystem Services in a Changing Climate' (BECC) funded by the Swedish government. M.P.C. and M.L. worked under the European Union Reference Laboratory (EURL) for Bee Health mandate.

**Author contributions** C.N., J.K. and M.R. conceived this study. A.K., A.-M.K., C.C., C.D., E.A., G. DiP., I.B., J.C.S., J.R.d.M., K.I., M.A., M.J.F.B., M.L., M.M., M.-P.C., M.R., O.S., P.M., P.D.l.R., R.R., S.G.P., S.H., T.B. and T.K. undertook the design and methodology. A.K., A.-M.K., C.C., C.D., D.S., E.C., G.d.P., G.T., M.H.P.-P., I.B., J.K., J.M., J.S., M.A., M.L., M.-P.C., M.R., O.S., P.M., R.R., S.G.P., S.H., T.K. and V.M.-L. acquired the data. C.N., C.C., C.D., O.S., T.K. and V.K. were involved in data analysis. C.N., C.C., C.D., J.K., M.R., O.S. and T.K. were involved in data interpretation. C.N., C.C., J.K., M.R. and T.K. wrote the original draft. All authors contributed to reviewing and editing the final manuscript.

**Funding** Open access funding provided by Lund University.

**Competing interests** The authors declare no competing interests.

**Additional information**
**Correspondence and requests for materials** should be addressed to Charlie C. Nicholson, Jessica Knapp or Maj Rundlöf.

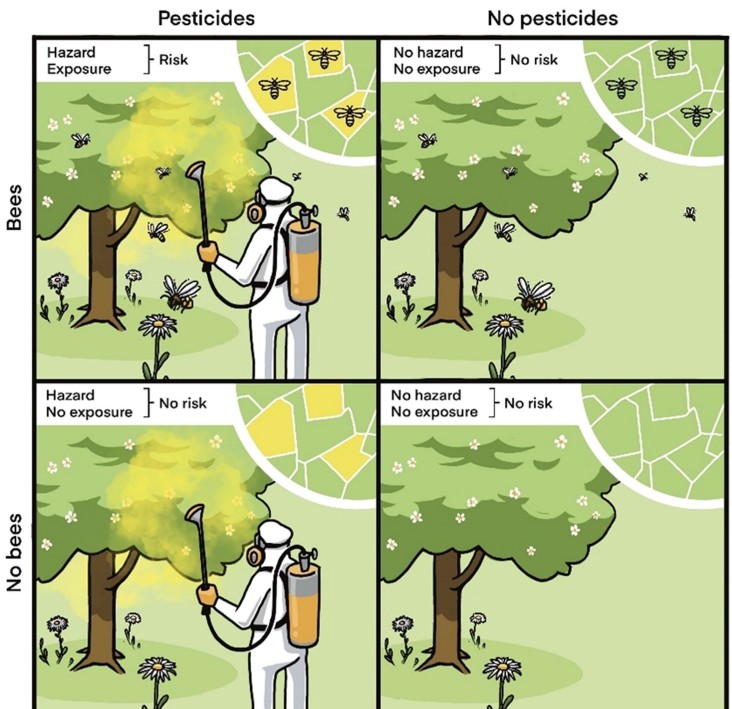

**Extended Data Fig. 1 | A simplified view of landscape exposure and resulting pesticide risk to bees.** Pesticide use creates potential hazard for non-target organisms. For bees in agricultural landscapes, pesticide risk results when their activity exposes them to this hazard (top left panel). Without the co-occurrence of hazard and exposure we expect no risk (remaining panels). Of course, the degree of hazard and exposure will depend on pesticide properties (e.g., toxicity, environmental fate, product formulations, use patterns) and bee traits (e.g., foraging range, sociality, body size, detoxification pathways). Moreover, real-world exposure occurs at landscape scales (see insets), because bees can integrate multiple sources of exposure by visiting spatially separated patches that vary in the identity, amount, timing and toxicity of hazard. We use the colony pollen stores collected by bumble bees (*Bombus terrestris*) to quantify pesticide risk resulting from this landscape exposure. We quantify exposure as the concentrations (μg/kg) of 267 substances in the pollen while hazard is quantified by the substances' toxicities ($LD_{50s}$). Scaling concentrations by toxicities and summing these toxicity-weighted concentrations provides a relative measure of pesticide risk to bees.

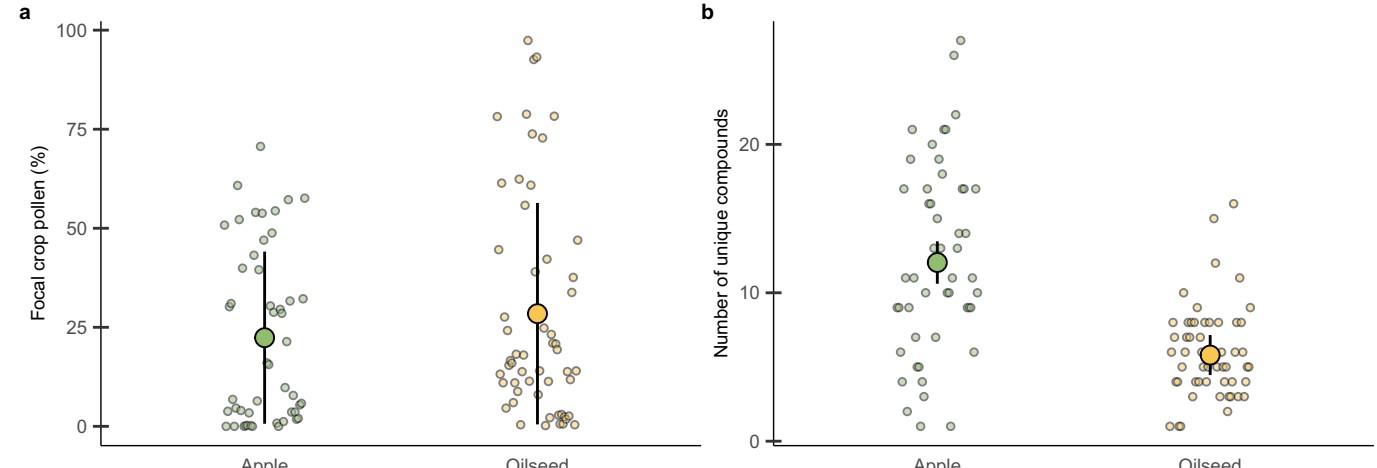

**Extended Data Fig. 2 | Percent of focal crop pollen and the number of pesticide compounds in pollen stores.** Bumble bee colony stores contained a substantial but variable portion of focal crop pollen types (**a**). More pesticide compounds were found in pollen collected at apple ($n$ = 50) sites than at oilseed rape ($n$ = 56) sites (**b**, $F_{1,104}$ = 39.59, $P$ < 0.001). For proportion of focal crop pollen (**a**), large points are means based on raw data and error bars are standard deviations. For the number of unique compounds (**b**), large points are estimated means from linear models and error bars are 95% confidence intervals. Small points are the individual data points.

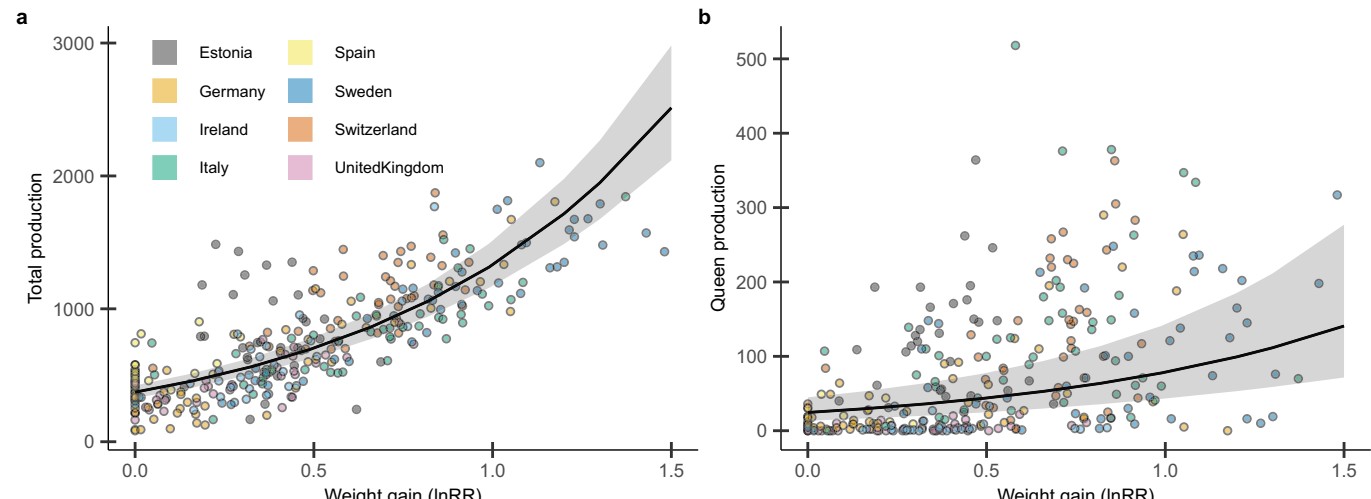

**Extended Data Fig. 3 | Bumble bee colony performance metrics are correlated.** Colony weight gain (response ratio: $\ln(g_{max}/g_{initial})$) is positively related with (**a**) total colony production (total number of produced bees estimated by the sum of closed and eclosed cocoons; $\chi^2 = 354.27$, $P < 0.001$) and (**b**) queen production (sum of closed and eclosed queen cocoons; $\chi^2 = 37.42$, $P < 0.001$). Fitted lines are estimated based on generalized linear mixed effects models with a negative binomial error distribution. Shaded areas represent the regression 95% confidence intervals. Point colours correspond to country colours in Fig. 1a and Extended Data Fig. 5a.

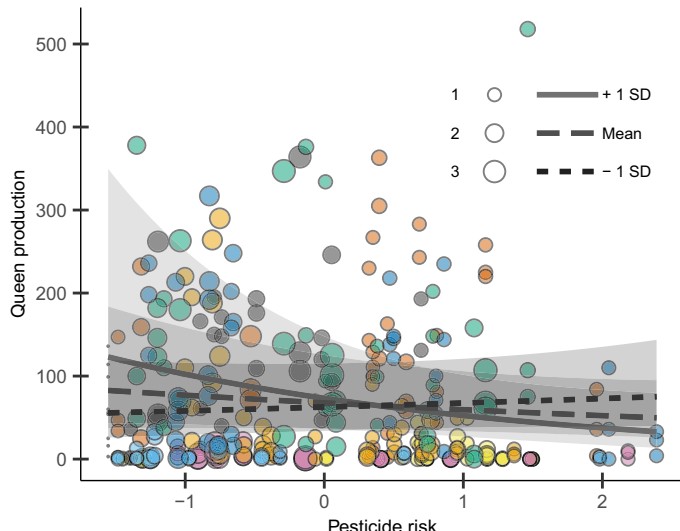

**Extended Data Fig. 4 | Effects of field exposure to pesticides on bumble bee colony queen production.** Colony queen production (sum of closed and eclosed queen cocoons) declined with pesticide risk (centred toxicity-weighted pesticide concentrations in pollen stores, see Methods) and landscape proportion cropland modified this effect, with stronger declines in landscapes with more cropland (solid line +1 *SD* proportion of cropland). Points are scaled by pesticide maximum cumulative ratio (MCR), the factor by which the mixture of compounds is riskier than the single most risky compound (see Methods). Fitted lines are estimated based on generalized linear mixed effects models with a negative binomial error distribution. Shaded areas and error bars represent the regression 95% confidence intervals. Results from statistical models are given in Extended Data Table 1. Point colours correspond to country colours in Fig. 1a and Extended Data Fig. 5a.

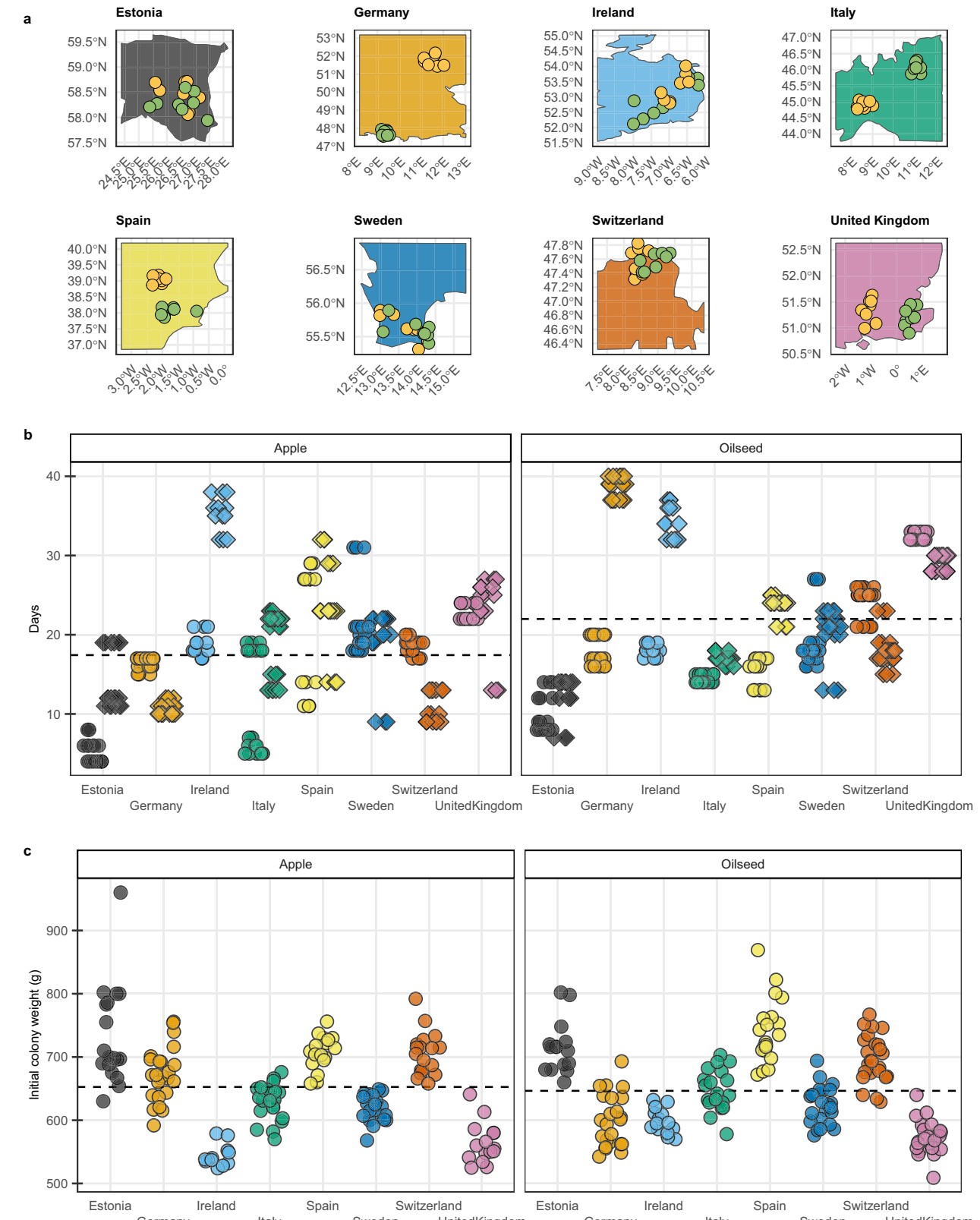

**Extended Data Fig. 5 | Overview of study design and set-up.** Our research network (**a**) included 128 sites in two focal crops (apple: green points, oilseed rape: yellow points) in eight European countries. At each site, three bumble bee (*Bombus terrestris*) colonies were deployed prior to focal crop bloom and weighed three times: before, during and after focal crop bloom. (**b**) The interval between first and second weights (circles) and second and third weights (diamonds) varied depending on region- and crop-specific bloom periods. (**c**) Colony weights at the time of deployment. Crop averages for number of days (**a**) and initial weight (**b**) across colonies are given as dashed lines. Site coordinates (**a**) are randomly jittered to protect farmer confidentiality. Colours in **b** and **c** correspond to country colours in **a** and Fig. 1a.

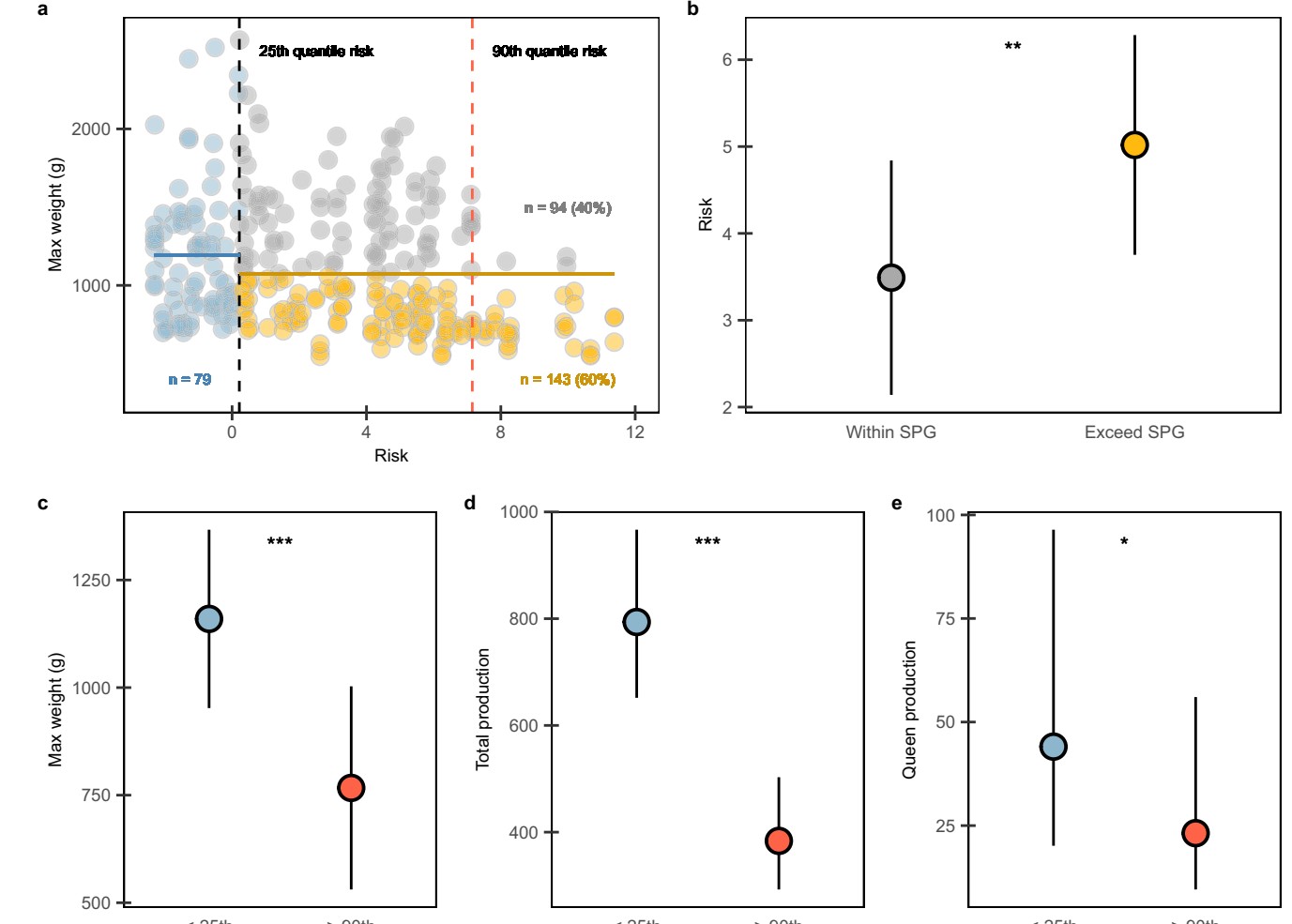

**Extended Data Fig. 6 | Pesticide risk effects exceed a suggested Specific Protection Goal (SPG).** To evaluate the magnitude of pesticide risk for the bees (**a**), we assume that the colonies belonging to a low-risk group (blue points, 25th percentile of risk) can be used to calculate the average maximum weight as a baseline (blue line). Using a suggested SPG for bumble bees of 10% reduction in colony weight[42] (yellow line), 60% of the remaining colonies in our study exceed this. In (**b**) we compare the risk for colonies that exceed the SPG (yellow points; n = 143 colonies) to those that do not (grey points; n = 94). The SPG is meant to protect 90% of the colony population across Europe, therefore in (**c-e**) we compare colony performance endpoints between the baseline colony group (blue, 25th percentile of risk; n = 79) and a high-risk colony group (red, 90th percentile of risk; n = 30) based on (**a**). Points and error bars (**b**–**e**) depict estimated means and 95% confidence intervals from linear (**b,c**) and generalized linear mixed effects models (**d,e**). * P < 0.05, ** P < 0.01 and *** P < 0.001. Exact p-values from statistical models are given in Extended Data Table 2.

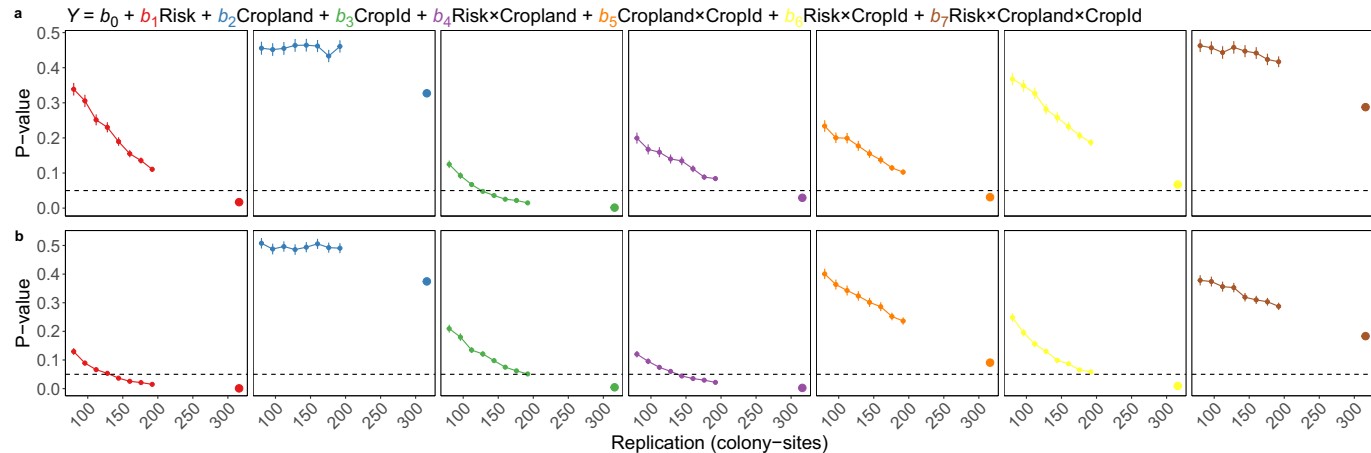

**Extended Data Fig. 7 | Sensitivity analysis of mixed effect models for the effect of pesticide risk, focal crop and proportion cropland.** We specified the same model structure (equation above plots) and simulated different levels of replication through stratified subsampling without replacement of colonies at the focal crop-country level and repeated the analyses for weight gain (**a**) and production (**b**) with these rarefied data sets. Levels of replication spanned 5 colonies per focal crop and country ($N = 80$ colonies) to 12 ($N = 192$), with this later value being the lowest focal crop-country level of replication in the data. The large point depicts the level of replication ($N = 316$ colonies) and p-values reported in the main text (See Table 1). Points and error bars (**a**,**b**) are means and 95% confidence intervals calculated over 1000 iterations per replication level.

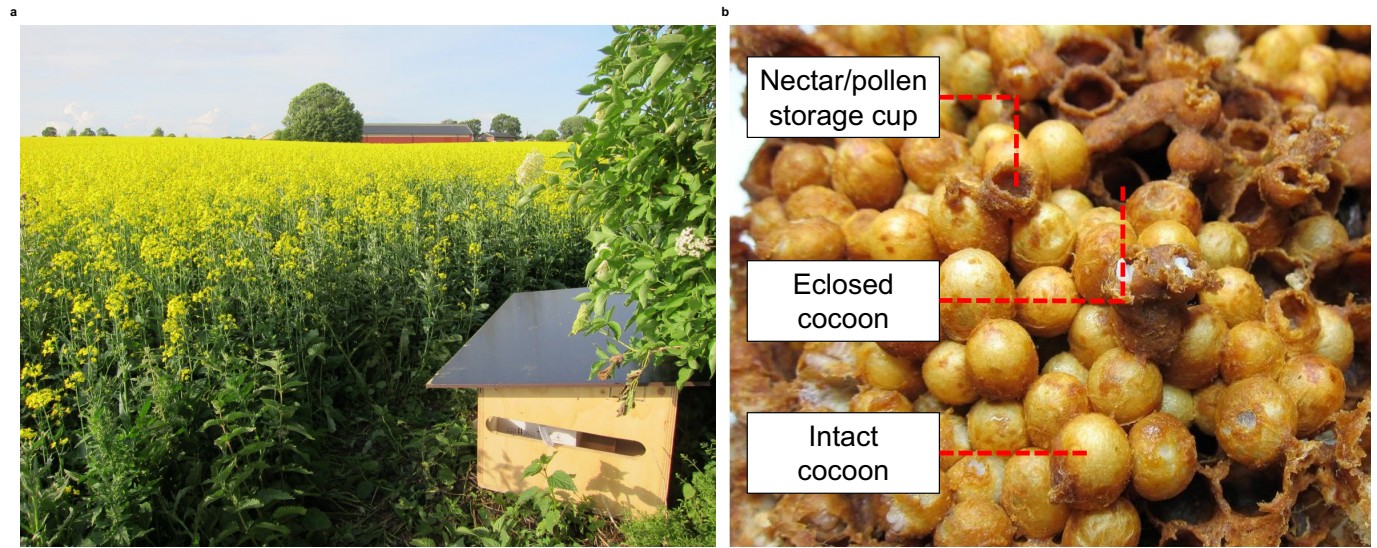

**Extended Data Fig. 8 | *Bombus terrestris* colonies in the field (a) and under dissection in the laboratory (b).**

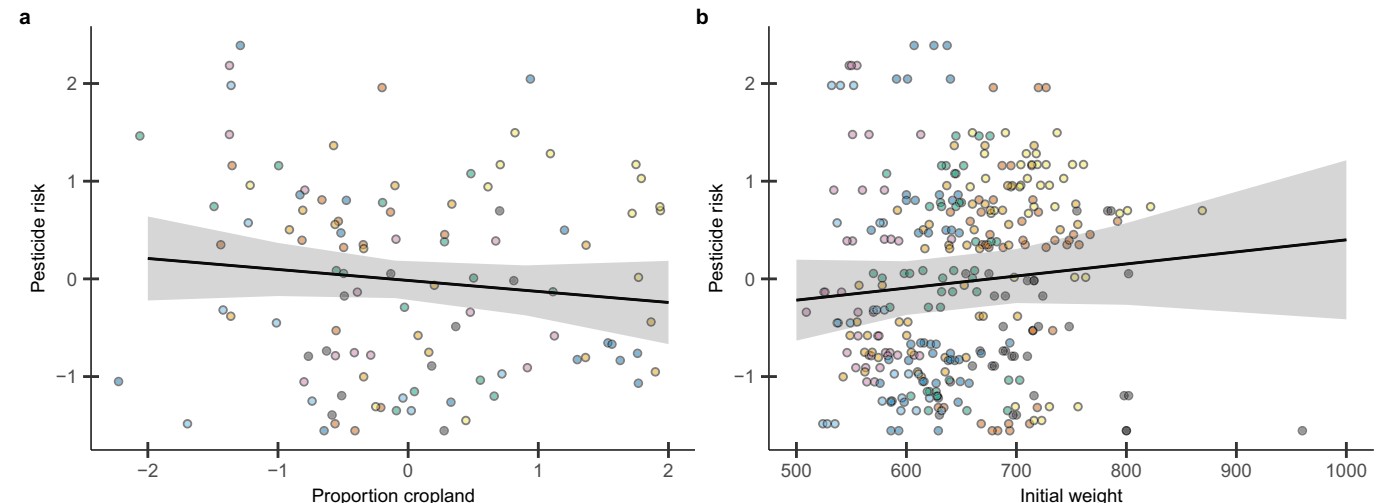

**Extended Data Fig. 9 | Risk is not correlated with (a) proportion cropland or (b) initial colony weight.** Fitted lines and 95% confidence intervals (**a,b**) are estimated based on linear mixed effects models. Point colours correspond to country colours in Fig. 1a, S5a and Extended Data Fig. 5a.

**Extended Data Table 1 | Additional colony performance results**

| | log Maximum weight | | Maximum weight | | Queen cocoon production | |
|---|---|---|---|---|---|---|
| | $\chi^2$ | $P$ | $\chi^2$ | $P$ | $\chi^2$ | $P$ |
| Initial weight | 0.26 | 0.61 | 0.00 | 0.99 | 5.58 | **0.018** |
| Risk | 10.60 | **0.0011** | 8.88 | **0.0029** | 0.25 | 0.61 |
| Crop | 8.41 | **0.0037** | 6.54 | **0.0105** | 5.39 | **0.0203** |
| Cropland | 1.26 | 0.26 | 1.82 | 0.17 | 0.12 | 0.73 |
| Risk × Crop | 8.78 | **0.0031** | 8.95 | **0.0028** | 0.61 | 0.44 |
| Risk × Cropland | 10.48 | **0.0012** | 12.18 | **0.0005** | 10.29 | **0.0013** |
| Crop × Cropland | 3.51 | 0.061 | 3.64 | 0.056 | 3.49 | 0.062 |
| Risk × Crop × Cropland | 0.66 | 0.42 | 0.96 | 0.327 | 1.17 | 0.28 |

Colony log-transformed maximum weight, maximum weight and queen cocoon production in relation to initial colony weight, pesticide risk (sum of toxicity-weighted pesticide concentrations: see Methods), crop type (oilseed rape, apple), the proportion of cropland in the surrounding landscape (1 km radius) and their interactions.

**Extended Data Table 2 | Comparisons of colony performance between risk groups**

| Response | $\chi^2$ | $P$ |
|---|---|---|
| Risk | 6.85 | **0.0088** |
| Colony maximum weight | 15.793 | **0.00007** |
| Total Production | 25.91 | **0.00004** |
| Queen Production | 6.33 | **0.012** |

Colony risk between colonies' SPG status and maximum weight, total production and queen production between low-risk (25th precentile of risk) and high-risk (90th percentile of risk) colony groups. Results are from linear (risk and weight) and generalized linear mixed effects models. See Extended Data Fig. 6 for details on SPG and risk groups.

# Reporting Summary

## Statistics

For all statistical analyses, confirm that the following items are present in the figure legend, table legend, main text, or Methods section.

| n/a | Confirmed | |
|---|---|---|
| ☐ | ☒ | The exact sample size (*n*) for each experimental group/condition, given as a discrete number and unit of measurement |
| ☐ | ☒ | A statement on whether measurements were taken from distinct samples or whether the same sample was measured repeatedly |
| ☐ | ☒ | The statistical test(s) used AND whether they are one- or two-sided *Only common tests should be described solely by name; describe more complex techniques in the Methods section.* |
| ☐ | ☒ | A description of all covariates tested |
| ☐ | ☒ | A description of any assumptions or corrections, such as tests of normality and adjustment for multiple comparisons |
| ☐ | ☒ | A full description of the statistical parameters including central tendency (e.g. means) or other basic estimates (e.g. regression coefficient) AND variation (e.g. standard deviation) or associated estimates of uncertainty (e.g. confidence intervals) |
| ☐ | ☒ | For null hypothesis testing, the test statistic (e.g. *F*, *t*, *r*) with confidence intervals, effect sizes, degrees of freedom and *P* value noted *Give P values as exact values whenever suitable.* |
| ☒ | ☐ | For Bayesian analysis, information on the choice of priors and Markov chain Monte Carlo settings |
| ☐ | ☒ | For hierarchical and complex designs, identification of the appropriate level for tests and full reporting of outcomes |
| ☒ | ☐ | Estimates of effect sizes (e.g. Cohen's *d*, Pearson's *r*), indicating how they were calculated |

*Our web collection on statistics for biologists contains articles on many of the points above.*

## Software and code

Policy information about availability of computer code

| Data collection | No software was used in the collection of bumblebee colony performance metrics.<br><br>For the detection of pesticide residues in pollen Analyst 1.6.2 software was used to control LC– MS/MS system and for data acquisition. Quantitative and qualitative analysis was done with MultiQuant software version 3.0 . Mass Hunter software version B.07.01 to control GC-MS/MS system and for data acquisition. Quantitative and qualitative analysis was done also with Mass Hunter software. For more information, see Kiljanek et al. 2021, Talanta (https://doi.org/10.1016/j.talanta.2021.122721)<br><br>GIS were made with ArcGIS Pro 2.4.1, ESRI. |
|---|---|
| Data analysis | All data were analysed in R (version 4.1.1.) using the following packages:<br>> linear mixed effects models, 'lme4' (Bates D, Mächler M, Bolker B 2015; v. 1.1.27.1)<br>> general linear mixed effects models, 'glmmTMB' (Brooks et al. 2017; v. 1.1.3)<br>> estimated marginal means, 'emmeans' (Lenth 2022; v. 1.7.2)<br>> interaction slopes, 'interactions' (Long 2019; v. 1.1.0)<br>> model diagnostics, 'performance' (Lüdecke et al. 2021; v. 0.10.4) and 'DHARMA' (Hartig 2021; v. 0.4.5) |

For manuscripts utilizing custom algorithms or software that are central to the research but not yet described in published literature, software must be made available to editors and reviewers. We strongly encourage code deposition in a community repository (e.g. GitHub). See the Nature Portfolio guidelines for submitting code & software for further information.

## Data

Policy information about availability of data

All manuscripts must include a data availability statement. This statement should provide the following information, where applicable:

- Accession codes, unique identifiers, or web links for publicly available datasets
- A description of any restrictions on data availability
- For clinical datasets or third party data, please ensure that the statement adheres to our policy

> The datasets analysed for the current study are available via the PoshBee project (Deliverable D1.6 Database of field records) and through FigShare: 10.6084/m9.figshare.24235573. Individual toxic endpoints (LD50s) for pesticide active ingredients came from the Pesticide Properties Database (http://sitem.herts.ac.uk/aeru/ppdb/)

## Human research participants

Policy information about studies involving human research participants and Sex and Gender in Research.

| | |
|---|---|
| Reporting on sex and gender | No human subjects were used in this research |
| Population characteristics | No human subjects were used in this research |
| Recruitment | No human subjects were used in this research |
| Ethics oversight | No human subjects were used in this research |

Note that full information on the approval of the study protocol must also be provided in the manuscript.

# Field-specific reporting

Please select the one below that is the best fit for your research. If you are not sure, read the appropriate sections before making your selection.

☐ Life sciences   ☐ Behavioural & social sciences   ☒ Ecological, evolutionary & environmental sciences

For a reference copy of the document with all sections, see nature.com/documents/nr-reporting-summary-flat.pdf

# Life sciences study design

All studies must disclose on these points even when the disclosure is negative.

| | |
|---|---|
| Sample size | Describe how sample size was determined, detailing any statistical methods used to predetermine sample size OR if no sample-size calculation was performed, describe how sample sizes were chosen and provide a rationale for why these sample sizes are sufficient. |
| Data exclusions | Describe any data exclusions. If no data were excluded from the analyses, state so OR if data were excluded, describe the exclusions and the rationale behind them, indicating whether exclusion criteria were pre-established. |
| Replication | Describe the measures taken to verify the reproducibility of the experimental findings. If all attempts at replication were successful, confirm this OR if there are any findings that were not replicated or cannot be reproduced, note this and describe why. |
| Randomization | Describe how samples/organisms/participants were allocated into experimental groups. If allocation was not random, describe how covariates were controlled OR if this is not relevant to your study, explain why. |
| Blinding | Describe whether the investigators were blinded to group allocation during data collection and/or analysis. If blinding was not possible, describe why OR explain why blinding was not relevant to your study. |

# Behavioural & social sciences study design

All studies must disclose on these points even when the disclosure is negative.

| | |
|---|---|
| Study description | Briefly describe the study type including whether data are quantitative, qualitative, or mixed-methods (e.g. qualitative cross-sectional, quantitative experimental, mixed-methods case study). |
| Research sample | State the research sample (e.g. Harvard university undergraduates, villagers in rural India) and provide relevant demographic |

| Research sample | *information (e.g. age, sex) and indicate whether the sample is representative. Provide a rationale for the study sample chosen. For studies involving existing datasets, please describe the dataset and source.* |
| --- | --- |
| Sampling strategy | *Describe the sampling procedure (e.g. random, snowball, stratified, convenience). Describe the statistical methods that were used to predetermine sample size OR if no sample-size calculation was performed, describe how sample sizes were chosen and provide a rationale for why these sample sizes are sufficient. For qualitative data, please indicate whether data saturation was considered, and what criteria were used to decide that no further sampling was needed.* |
| Data collection | *Provide details about the data collection procedure, including the instruments or devices used to record the data (e.g. pen and paper, computer, eye tracker, video or audio equipment) whether anyone was present besides the participant(s) and the researcher, and whether the researcher was blind to experimental condition and/or the study hypothesis during data collection.* |
| Timing | *Indicate the start and stop dates of data collection. If there is a gap between collection periods, state the dates for each sample cohort.* |
| Data exclusions | *If no data were excluded from the analyses, state so OR if data were excluded, provide the exact number of exclusions and the rationale behind them, indicating whether exclusion criteria were pre-established.* |
| Non-participation | *State how many participants dropped out/declined participation and the reason(s) given OR provide response rate OR state that no participants dropped out/declined participation.* |
| Randomization | *If participants were not allocated into experimental groups, state so OR describe how participants were allocated to groups, and if allocation was not random, describe how covariates were controlled.* |

# Ecological, evolutionary & environmental sciences study design

All studies must disclose on these points even when the disclosure is negative.

| Study description | In each of eight European countries (Estonia, Germany, Ireland, Italy, Spain, Sweden, Switzerland, United Kingdom) we selected 16 sites split evenly between two focal crops (Apple: N = 64, Oilseed: N = 64) following a predefined PoshBee protocol (Schweiger et al. 2019). All sites were > 3 km apart to ensure spatial independence of bumblebee foraging ranges. Sites were selected to represent a gradient of agrochemical use and proportion of cropland. in the surrounding landscape The proportion of cropland coverage ranged from 3-98% within 1-km radius buffers. See Hodge et al. (2022) for setup of the study system using a multi-actor approach.

Schweiger, O., Hodge, S., Rundlöf, M., Dominik, C. (2019) WP1.1.1 Field site selection. PoshBee protocol.

Hodge, S., Schweiger, O., Klein, A. M., Potts, S. G., Costa, C., Albrecht, M., ... & Stout, J. C. (2022). Design and planning of a transdisciplinary investigation into farmland pollinators: rationale, co-design, and lessons learned. Sustainability, 14(17), 10549. |
| --- | --- |
| Research sample | At each site we placed three colonies (N = 384) of Bombus terrestris terrestris, except in Ireland and the United Kingdom where the local subspecies Bombus terrestris audax was used. Colonies were all sourced through local providers. Before deployment colonies were checked for a natal queen and weighed (648 ± 70.9 g, mean ± SD). Colonies were deployed prior to crop bloom. Colonies were housed in protective structures. From each of these colonies we collected 1) metrics of colony performance and 2) pollen samples for pesticide residue analysis and palynological identification.

For metrics of colony performance we determined the weight change and maximum weight achieved by individual colonies and colony total production. For weight, we measured each colony before, during, and after bloom and took the maximum value. For weight change, we calculated the natural-log response ratio for each colony as ln(g_max/g_initial), where g_initial is weight prior to bloom and g_max is the maximum weight achieved by a colony during its field placement. For colony production, we closed colonies after bloom and froze them (-20 C) for laboratory dissection. From dissected colonies we counted the number of intact and eclosed worker/male and queen cocoons, including the eclosed cocoons used for nectar and pollen storage. We sum these colony structures because individual counts may be unrepresentative due to absent colony members at the time of retrieval.

After dissection, we extracted pollen stored in the colonies and pooled pollen evenly from all three colonies, aiming to obtain at least 15.0 g. Pollen samples were stored below -20 C. Samples were sent on dry ice to PIWet, where each site sample was homogenized and split for the analysis of pollen identity (1/5 sample amount) and pesticides residues (remaining sample amount). Analyses of residues were performed for samples in which quantity was at least 0.52 g (see Data Exclusions).

For pesticide residue analysis, we used a previously described method (Kiljanek et al., 2021) that is validated according to SANTE/12682/2019 (European Commission et al., 2020) and accredited in accordance with the ISO 17025 standard. Reagent blanks and blank samples were analyzed in each batch. Recovery checks with samples spiked with pesticides at LOQ levels were performed in each analytical batch to meet SANTE/12682/2019 criteria.

For pollen identification, palynological analyses were performed at the Research Centre for Agriculture and Environment (CREA) Bologna, Italy, a laboratory specialized in analyses of bees and bee products and accredited according to UNI CEI EN ISO/IEC 17025. Recognition of pollen type was based on comparison between the observed pollen forms and those present in the CREA collection of reference slides (developed using anthers of identified plant species). For each pollen type, the percentage with respect to the total number of counted pollen grains was calculated.

Land use data is based on high resolution images provided by World Imagery (ESRI) land cover features were classified at a consistent scale of 1:2500. World Imagery provides one meter satellite and aerial imagery, typically within 3-5 years of currency, using a combination of imagery sources such as 2.5m SPOT imagery and 0.5m resolution imagery from DigitalGlobe. |

While performance metrics are replicated at the colony level, pesticide, pollen and landscape data are replicated at the site level. To accommodate this hierarchical data structure we used mixed effects models with site nested within country as a random factor

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

| Sampling strategy | For site and colony sampling, we based our replication on previous power analyses performed by co-authors (EFSA 2022; Woodcock et al. 2016).<br><br>References:<br>European Food Safety Authority (EFSA), Auteri, D., Arce, A., Ingels, B., Marchesi, M., Neri, F.M., Rundlöf, M. and Wassenberg, J., 2022. Analysis of the evidence to support the definition of Specific Protection Goals for bumble bees and solitary bees (Vol. 19, No. 1, p. 7125E).<br><br>Woodcock, B. A., Heard, M. S., Jitlal, M. S., Rundlöf, M., Bullock, J. M., Shore, R. F., & Pywell, R. F. (2016). Replication, effect sizes and identifying the biological impacts of pesticides on bees under field conditions. Journal of Applied Ecology, 1358-1362. |
|---|---|
| Data collection | For colony performance, data was collected both in the field and in the lab by PosheBee project members and associated technicians following a predefined protocol (Rundlöf & Hodge 2019).<br><br>For pesticide residues, analyses were performed at the Department of Pharmacology and Toxicology, National Veterinary Research Institute, Puławy, which is the National Reference Laboratory for pesticide residue analysis and regularly participates in international proficiency tests with satisfactory results. Data on pesticide properties and toxicity information (used in the calculation of pesticide risk) were extracted from the Pesticide Properties Database (PPDB) hosted by the University of Hertfordshire.<br><br>For land use, all landscape features were identified and mapped within a 1-km radius, centred at the location of the sentinel bee colonies, using heads-up digitizing in a geographic information system (ArcGIS Pro 2.4.1, ESRI) and EUNIS habitat classification as a guideline. Data collection was performed by the PoshBee WorkPackage 1-2 teams.<br><br>All data collection adhered to a pre-formulated data management plan for the PoshBee project.<br><br>Rundlöf, M., Hodge, S. (2019) WP1.5.9 Assessment of Bombus terrestris colony performance and natural enemies. PoshBee protocol. |
| Timing and spatial scale | Colonies were deployed between 2019-03-24 and 2019-05-12 depending on region and crop specific bloom times. Similarly, colonies were retrieved between 2019-05-06 and 2019-06-12. Colonies were weighed three times i) before installment at the field sites, ii) during the middle of flowering, iii) when the colonies are retrieved from the field sites. Total production, pesticide residue and pollen identity data collection happened after colonies were retrieved from the field.<br><br>The spatial scale of the study encompasses eight European countries (see map figure in manuscript). We analyzed 106 circular landscapes with 1km radius, thus a total of 332.84 km^2 were considered. |
| Data exclusions | Of the 384 colonies initially deployed across 128 sites (64 apple, 64 oilseed rape), we analysed 316 colonies from 106 sites. This reduced sample size is due to colony losses (e.g., raccoon attack, overrun by machinery) or colonies not yielding enough pollen store material for pesticide quantification (N = 5 for loss, N = 63 for insufficient pollen for pesticide analyses). See Extended Data Table 3 |
| Reproducibility | No attempts were made to repeat the full study but it was to some extent built into the experimental design by including eight countries. |
| Randomization | No randomization was used. |
| Blinding | No explicit blinding was used. However, field data collectors were not aware of the pesticide use in the focal field or the surrounding landscape, or in the pollen stores, since this data was collected through laboratory analysis after the field part of the study was completed. Conversely, laboratory data collectors were generally not aware of the colony performance measure when the pesticide residue data and pollen identity data were collected. |

Did the study involve field work?   ☒ Yes   ☐ No

# Field work, collection and transport

| Field conditions | Agricultural landscapes. We chose oilseed rape and apple as our focal crops to reflect annual and perennial cropping practices and, therefore, different pest pressures, pest management strategies and pesticide use. The most dominant land covers were cropland (mean: 55%, range: 3-98%) and semi-natural habitat (mean: 37%, range: 0.1-93%), which where the latter was comprised of grasslands (mean: 19%, range: 0.1-76%), woodlands (mean: 18%, range: 0-62%), and wetlands (mean: 0.1%, range: 0-3%). |
|---|---|
| Location | Estonia, Germany, Ireland, Italy, Spain, Sweden, Switzerland, United Kingdom |

| Access & import/export | Access to farm locations was granted by property owners. Bumble bee colonies were sourced through local providers which held import permits. |
|---|---|
| Disturbance | No disturbance was caused by the study. |

# Reporting for specific materials, systems and methods

We require information from authors about some types of materials, experimental systems and methods used in many studies. Here, indicate whether each material, system or method listed is relevant to your study. If you are not sure if a list item applies to your research, read the appropriate section before selecting a response.

### Materials & experimental systems

| n/a | Involved in the study |
|---|---|
| ☐ ☐ | Antibodies |
| ☐ ☐ | Eukaryotic cell lines |
| ☐ ☐ | Palaeontology and archaeology |
| ☐ ☒ | Animals and other organisms |
| ☐ ☐ | Clinical data |
| ☐ ☐ | Dual use research of concern |

### Methods

| n/a | Involved in the study |
|---|---|
| ☒ ☐ | ChIP-seq |
| ☒ ☐ | Flow cytometry |
| ☒ ☐ | MRI-based neuroimaging |

## Antibodies

| Antibodies used | *Describe all antibodies used in the study; as applicable, provide supplier name, catalog number, clone name, and lot number.* |
|---|---|
| Validation | *Describe the validation of each primary antibody for the species and application, noting any validation statements on the manufacturer's website, relevant citations, antibody profiles in online databases, or data provided in the manuscript.* |

## Eukaryotic cell lines

Policy information about cell lines and Sex and Gender in Research

| Cell line source(s) | *State the source of each cell line used and the sex of all primary cell lines and cells derived from human participants or vertebrate models.* |
|---|---|
| Authentication | *Describe the authentication procedures for each cell line used OR declare that none of the cell lines used were authenticated.* |
| Mycoplasma contamination | *Confirm that all cell lines tested negative for mycoplasma contamination OR describe the results of the testing for mycoplasma contamination OR declare that the cell lines were not tested for mycoplasma contamination.* |
| Commonly misidentified lines (See ICLAC register) | *Name any commonly misidentified cell lines used in the study and provide a rationale for their use.* |

## Palaeontology and Archaeology

| Specimen provenance | *Provide provenance information for specimens and describe permits that were obtained for the work (including the name of the issuing authority, the date of issue, and any identifying information). Permits should encompass collection and, where applicable, export.* |
|---|---|
| Specimen deposition | *Indicate where the specimens have been deposited to permit free access by other researchers.* |
| Dating methods | *If new dates are provided, describe how they were obtained (e.g. collection, storage, sample pretreatment and measurement), where they were obtained (i.e. lab name), the calibration program and the protocol for quality assurance OR state that no new dates are provided.* |

☐ Tick this box to confirm that the raw and calibrated dates are available in the paper or in Supplementary Information.

| Ethics oversight | *Identify the organization(s) that approved or provided guidance on the study protocol, OR state that no ethical approval or guidance was required and explain why not.* |
|---|---|

Note that full information on the approval of the study protocol must also be provided in the manuscript.

# Animals and other research organisms

Policy information about [studies involving animals](); [ARRIVE guidelines]() recommended for reporting animal research, and [Sex and Gender in Research]()

| | |
|---|---|
| Laboratory animals | Bombus terrestris terrestris and Bombus terrestris audax colonies. The colonies were approximately 10 weeks old and contained one queen, approximately 100 workers and brood in all stages. |
| Wild animals | This study did not involve wild animals |
| Reporting on sex | *Indicate if findings apply to only one sex; describe whether sex was considered in study design, methods used for assigning sex. Provide data disaggregated for sex where this information has been collected in the source data as appropriate; provide overall numbers in this Reporting Summary. Please state if this information has not been collected. Report sex-based analyses where performed, justify reasons for lack of sex-based analysis.* |
| Field-collected samples | Colonies were housed in protective structures to limit weather and predator exposure. Before deployment, we confirmed that each colony had a natal queen and recorded its initial weight. No colony maintenance (e.g., supplemental feeding) was performed. At the end of the experiment colonies were weighed again, then sealed, transported from sites, and stored at -20 C. |
| Ethics oversight | No ethical approval was required because this study involved experimentally placed insect invertebrates. |

Note that full information on the approval of the study protocol must also be provided in the manuscript.

# Clinical data

Policy information about [clinical studies]()
All manuscripts should comply with the ICMJE [guidelines for publication of clinical research]() and a completed [CONSORT checklist]() must be included with all submissions.

| | |
|---|---|
| Clinical trial registration | *Provide the trial registration number from ClinicalTrials.gov or an equivalent agency.* |
| Study protocol | *Note where the full trial protocol can be accessed OR if not available, explain why.* |
| Data collection | *Describe the settings and locales of data collection, noting the time periods of recruitment and data collection.* |
| Outcomes | *Describe how you pre-defined primary and secondary outcome measures and how you assessed these measures.* |

# Dual use research of concern

Policy information about [dual use research of concern]()

## Hazards

Could the accidental, deliberate or reckless misuse of agents or technologies generated in the work, or the application of information presented in the manuscript, pose a threat to:

| No | Yes | |
|---|---|---|
| ☒ | ☐ | Public health |
| ☒ | ☐ | National security |
| ☒ | ☐ | Crops and/or livestock |
| ☒ | ☐ | Ecosystems |
| ☒ | ☐ | Any other significant area |

## Experiments of concern

Does the work involve any of these experiments of concern:

| No | Yes | |
|---|---|---|
| ☒ | ☐ | Demonstrate how to render a vaccine ineffective |
| ☒ | ☐ | Confer resistance to therapeutically useful antibiotics or antiviral agents |
| ☒ | ☐ | Enhance the virulence of a pathogen or render a nonpathogen virulent |
| ☒ | ☐ | Increase transmissibility of a pathogen |
| ☒ | ☐ | Alter the host range of a pathogen |
| ☒ | ☐ | Enable evasion of diagnostic/detection modalities |
| ☒ | ☐ | Enable the weaponization of a biological agent or toxin |
| ☒ | ☐ | Any other potentially harmful combination of experiments and agents |

