## [Peer Review File · Nature]

Manuscript Title: Pesticide use negatively affects bumble bees across European landscapes

Reviewer Comments & Author Rebuttals

Reviewer Reports on the Initial Version:

Referees' comments:

Referee #1 (Remarks to the Author):

This manuscript describes a substantial and well-designed field experiment across 8 countries of the EU examining pesticide exposure of bumblebee colonies near apple and oilseed rape crops, and the associated impact on colony growth and production during crop bloom. The results clearly show that colonies are exposed to a vast array of pesticides, and there is a risk of this multiple exposure leading to harm at the colony level, and possibly at levels which fail to meet European Specific Protection Goals. The authors present this as strong evidence for post-approval monitoring of exposure and effects of pesticides.

Originality and significance:

The manuscript is novel and highly significant, and of interest across a wide field of researchers. It will also draw the attention of policy-makers at the forefront of crop and environmental protection.

Data & methodology:

The methods are robust and the data are technically sound; although it requires further explanation as to why the colonies were terminated when crop bloom finished and how the variability across countries and sites in this experimental time-frame could affect the results [see below in Suggested Improvements].

Statistics:

The statistical approach is thorough and the additional reporting summary is clear and in order.

Conclusions:

This was a large and well replicated experiment and the results are clear so I consider the conclusions robust and valid. I think the reflection on SPGs is useful; and the sensitivity analysis gives a wider perspective on the scale of testing required for effects to be detected.

Suggested improvements:

1) The intervals between colony weight measurements were based on crop bloom and varied a lot between sites (Fig S1c) – it looks like difference between g-initial and g-max varied between 10 days (Estonia, OSR) and 40 days. The authors do explain the caution required with interpretation (L106-115) but it would be helpful to do an analysis to test whether the experimental duration affected the results. Could 'days' be added as a co-variate? Or can we at least have a plot of days between g-initial and g-max versus total production?

2) In fact, for Fig S1c - why have you plotted days between 1st & 2nd weights and between 2nd and 3rd weights when you only analyse the weight change between the 1st and 3rd weights? It would be clearer to be able to see the number of days between 3rd (g-max) and 1st (g-initial) weights.

3) As colony termination was also timed with crop bloom rather than colony dynamics, then the measurement of colony production is only relevant to that period and not to the whole colony cycle. Whilst you have explained caution of interpretation, it would again be useful to have a

numeric demonstration of whether this experimental duration affects the actual results - in this case a plot of days between g-initial and g-max versus total production would be useful.

The writing style is a bit 'sloppy' in places which makes the reader have to read sentences twice before getting the exact meaning. Please go through and clarify as much as possible.

The colour-shading of the different countries in the figures is not very friendly to colour-blindness with lots of greenish and reddish tones. Can this be improved?

Minor revisions:

L57-58: this sentence needs unpacking a bit.

L119: Based on the experimental time windows it is more appropriate to describe bumblebee colony performance as "colony performance during crop bloom" or something to denote that it doesn't relate to the whole season.

Fig 1a: The scale doesn't allow reader to see relative location of apple and osr points for each country. Could you put 8 insets into the supplementary material showing distribution of these sites within each country more clearly.

Table 2: Title needs to say 'Ten compounds *found in the colony pollen stores* that pose most risk.....'. Otherwise it sounds like you might have got the data from somewhere else.

L527: Fig S1 legend: State that the x axis in plots c and d relate to the country. For clarity add '(c)' before 'The interval between.....'

References:

The manuscript is well referenced and set well within the context of this controversial field.

Abstract & Overall:

The abstract provides a very good summary of the work, and the conclusions are appropriate. The findings are important and of general interest.

Referee #2 (Remarks to the Author):

This paper describes field monitoring of bumblebee colonies (>300) across numerous agricultural locations (>100) in 8 European countries. The paper reports widescale prevalence of pesticide residues within the colonies and associates this with reduced viability of those colonies. I think this is a potentially important study, but I have some concerns over the interpretation (to an extent, I think what they have done is good). I have some issue with terminology (risk vs hazard, and that they don't really measure risk per se - however given the correlative nature of this I don't see this as a major issue, they just need to be more careful in their terminology). I think my major criticisms are: 1) You basically ignore queen production (you touch on it a bit) on the basis you sampled hives too early for this to be meaningful. This is such a major factor for understanding reproductive potential of hives and so pop. Persistence - you may show a reduction in hive weight but if this isn't having a major impact on queen / male production is this actually a risk. Certainly you could argue yes, but for me this would be a definitive argument. What you show may be a risk - that said I still think your findings are important; 2) You need to make more use of effect sizes somehow to indicate the scale of the effect - you do at one point but I am unclear how it was derived or what it compares to. I think comparing say effect size differences between your 25% lowest HQ colonies vs 25% highest Hq colonies, or relating it to 90th percentile HQ colonies to average would be valuable - your data has a lot of scatter and I want to know if this is an important or trivial difference. This is easy to do I suspect but may require some re-adjustment of interpretation. I think if the queen data was more robust it would be more of a slam-dunk for me, but I still think given the scale and scope of the study its Nature worthy - I just want more insight from you into the results as to whether the trends are of trivial importance or actually represent large effect sizes that affect major parts of the population.

Abstract

L37-39: not sure why you need to make such a big point about neonics here unless they are explicitly a major risk (which given the ban in the EU outside of some derogations seems less likely, but not impossible). I think its fair to say that risks to bees from insecticides (as well as other synergistic effects) are pretty likely (thus the regulatory process) so I don't see the need for a line saying neonics ore bad so other insecticides may also be bad.

L40: What do you mean by growth and development? Are you talking about individuals or a colony. I would argue that a lot of scientific studies (neonics to quote your example) impacts on growth and development of honeybees, bumblebees and Red Mason bees have been widely tested. Higher level Risk assessments also look at colony growth effects (see Pilling, E., et al (2013) A Four-Year Field Program Investigating Long-Term Effects of Repeated Exposure of Honey Bee Colonies to Flowering Crops Treated with Thiamethoxam. PLoS ONE, 8, e77193) and arguably lower tier work looking at chronic toxicity of honeybees and increasingly other species (although that's still limited) look at impacts on a range of lifestages from larvae to different casts. I kind of get your point but I think your overplaying this here - I think its the scale and consistency of your assessment that makes this exciting. Also you keep saying bees - what bee species are you talking about.

L42: I am finding the work associated a bit on the vague side. Is this that you have experimentally demonstrated reduced colony performance from exposure, or you have a list of pesticides and you are inferring potential impacts from a risk quotient or something like that, or your just saying there are pesticides in the hives and they might have an impact (I am guessing not the latter). Needs to be more specific.

Main text

L52: So that's also not true - there is a massive body of work looking at mixture effects and toxicity, from concentration addition models, TKTD models, NUTS models etc. Synergisms are defined as where they deviate from model predictions about additive toxicity of similar or different modes of action - see Cedergreen, N. (2014) Quantifying Synergy: A Systematic Review of Mixture Toxicity Studies within Environmental Toxicology. PLoS ONE, 9, e96580 for an example of this. All these models are intended to predict field level effects. Again I get what you are aiming for, that large scale experimental testing of mixture effects hasn't been done, but I also think your kind of ignoring the massive areas of ecotoxicology that does look at this to drive home your point. Specifically if you are looking at trying to show field level synergisms that these need to be relative to mortality effects you would expect from a some kind of mixture toxicity model - again see the ceadergreen paper.

L53-54: I don't think this statement is true - see points above. I can see what you are getting but there are multiple studies that look at this, including ones by authors on this paper.

L54-56: Although I think you need to be careful about the language and some of your statements, I do think this is important and its an impressive body of sites / colonies that underpins your assessment.

L58: not sure I like the 'nutritional benefit reduced by pesticide cost', again I kind of see what your saying in terms of a colony scale effect, but I think its a weird way of saying it. I also think its more likely to be relevant for chronic as opposed to acute toxicity effects and relative to the scale of exposure. Acute toxicity for a solitary species makes the effects of forage quality irrelevant. Also what about all the other drivers of population change like predation, disease, nesting site availability, landscape structure etc. Seems a bit of a simplified statement. That said statement in L58-60 seems reasonable as a working hypothesis.

L63-65: Sounds great - a really impressive sampling network.

L65: Can you do an overview of what active ingredients you are looking at. How many, are they just insecticides, or are there fungicides and herbicides. Does this cover both current licenced products or do you include out of licence products as well.

L66-71: this is a really important finding. When you say cropland moderated this effect do you mean this is because there is less crop (so less pesticides) or is this due to greater food resources from semi-natural. This could do with more explanation. Is this having an effect as an interaction, or is it just another covariate in the model. Its not clear from reading the text.

L69: so I guess one question I have is whether you had a look at Queen production. Given you could argue these are the reproductive units (see Rundlöf, et al (2015) Seed coating with a neonicotinoid insecticide negatively affects wild bees. *Nature*, 521, 77-80; Whitehorn, P.R., et al (2012) Neonicotinoid pesticide reduces Bumble Bee colony growth and queen production. *Science*, 336, 351-352) this seems an important thing to consider. At the end of the day if queen production is not negatively effected this is an important finding, even if colonies on average have lower weight gain or total number of cells. Without this you do have an important finding that certainly suggests there may be effects, but from a risk management perspective you could argue that if it doesn't impact on population viability (presumed from say queen production) is it a problem - probably yes it is although more subtle, and certainly you could argue it would be important for solitary species and for ecosystems service provision (where numbers of workers do matter), or for other bumblebee species that tend to have smaller total worker numbers to start with. I think you need queen production trends or if not an explanation that this may be important.

L66: Also general question is I think it would be useful to given net effect size changes in your key metrics over potentially: 1) the range of risk (or hazard quotients) you are finding in the hives; and 2) at a 90 percentile HQ values relative to the lowest HQ value to link in (kind of) with EFSA documentation on protecting 90% of the population / 90% exposure risk. This seems to be the protective standard they are going to so you may as well make use of this.

L73: So you do this repeatedly, you say cropland or crop moderated an effect - can you say how it moderated it. I appreciate you have limited space and a lot of results - but if it's a consistent overall trend in the way cropland had an effect maybe just describe this so its clear, but note there are exceptions.

L75: What do you mean by mor resource yielding - I am guessing you mean in terms of per area pollen / nectar. If this isn't something you tested (which I think it isn't as you have a reference) maybe save this statement for the discussion as it seems slightly out of place here.

L77: Ok you seem to have these overall summaries I asked for (I think) in the moderating effects of crop land. I would put these earlier and then give the details after. Or certainly this is a model for how to describe each moderating effect.

L81: Is it risk or hazard you are measuring. I.e. is it exposure (ng. bee-1) / toxicity (e.g. LD50 ng bee-1) or a measure of hazard where you don't have exposure per bee, but have a measure of residues concentrations. Its ecotoxicology what you are doing so I think you need to be careful about terminology.

L83: See earlier point about I want to know the range and types of pesticides you are assessing - just a few lines earlier would be good.

L86: Again is this a risk, or is this hazard. If you are dividing residues in pollen / toxicity you are not using the same units so its not really a risk, its just a scaled measure of hazard. Nothing wrong with that in the context of this study as its correlative looking at relative toxicity, but just be careful about language. Note I appreciate some HQ have been quantified to give a measure or risk (e.g. application to crop / LD50 has defined levels of concern, but those were experimentally validated. See Thompson, H.M. (2021) The use of the Hazard Quotient approach to assess the potential risk to honeybees (*Apis mellifera*) posed by pesticide residues detected in bee-relevant matrices is not appropriate. *Pest Management Science*, 77, 3934-3941. For a more detailed discussion of some of these issues).

L87: Most insecticides are bee health antagonists - they kill them with sufficient exposure. Are you talking about pesticides that may have synergistic interactions that results in a deviation in toxicity from predicted levels, or specifically impacts of some measure of bee health at an individual or colony level. Again I get what you are aiming for but the language needs tightening up.

L89: that needs more explanation in the main text. I am not sure what you are getting at here. I need to look at the methods maybe to get a better idea, but this seems a weird metric - also I have no idea what the range of 1- 3.8 relates to in terms of a Level of Concern (e.g. like a RQ > 1 for acute toxicity). Is it good or bad, or a unitless scale that just goes from presumed less bad to bad.

L99: Or just the widespread use of azole fungicides for arable farming could do the same.

L101: Agreed, however it could also overestimate it if there are antagonistic effects. Again see th4

cerdergreen paper. However, I am in general agreement with you that mixture effects risks are potentially hard to estimate. That said you could potentially test for this as you know where certain common synergistic products are found - e.g. azoles. You could include them as a covariate in your model as part of an interaction to see if where present you get greater than expected negative effects on your colony metrics. However, just a thought - I can see it's a bit of a throwaway line and not an unreasonably one so not asking you to reanalyse your data.

L102-103: Can you expand on this - its one of the its that sounding interesting but its not entirely clear what you are getting at.

L108-109: I think you need to have the males and queen reproduction to back up your argument as a response. You kind of touch on this in the next section but I think you need to go into more detail. Leaving it out is a major hole in your arguments. While colony weight can be correlated with queen numbers, I have seen papers where its not (I am going to be honest I cant remember which one). I think your arguemtyn holds, but it does undermine this by not having an explicit demonstration that queen production was reduced (and more importantly reduced by how much). This comes back to the effect size issue again. Ok yes you may have found a negative effect, but if from an effect size (as opposed to just comparing means and ignoring variance) perspective its 1-2% difference when comparing the best and worst case situations that's not that worrying, and sits within accepted thresholds. I guess this is the difference between whether what you report is unacceptable risk - 90% of colonies show a reduction in colony queen reproduction of >20% for example vs acceptable risks like 90% of colonies show a reduction in queen reproduction of <2%. This needs to be better qualified in the main paper text. It might not be as bad as it could be, but it would be scaled to be meaningful.

L130 - fantastic you have this - can you do more of this. Specifically for the queen production if you can. That said I don't get what you mean by using the average maximum colony weight across the colonies as reference. Are you saying when you compare to the worst possible case of pesticide hazard or risk based on residues there is a 10% reduction in colony size. This seems unquantified to me. What's the difference say compared to a 90th percentile worst case scenario of hazard relative to the mean. Its good though to have these figures and relate to a protection goal.

L132: Again this surely depends on what % of the population is reduced by 10% relative to the mean.

L135: Is this a post hoc power analysis. I don't get what your saying - maybe this would be better covered in the methods but acknowledge in the main text the need for lots or replication.

Specifically what I don't get with this statement is that by definition you had the power to detect the effects you saw (you have a p value to prove it) with the replication you had (which was 100 ish sites). Normally this would be to say you couldn't detct an effect size of say 5% with the replication you had so effects at that scale were undetectable. This may just be better in the methods where you can explain it a bit more as I feel I am missing something here.

L140-145: Overall I really agree with what you are saying. Lower tier reisk assessments focus on individual toxicity, not colony level effects so much, with risks here being the basis for kicking off higher tier assessments. - I guess a lot of this comes down to the failure of higher tier risk assessments that are quantifying exposure under field conditions (albeit probably more for honeybees). Although I guess on a practical level they focus on one active at a time, not field scale real world mixture exposure which is what you are quantifying. It seems to be more of a systemic failing of the regulatory approach to consider real world post regulation mixture 9additive or synergistic) toxicity.

Table 2: So means are inherently problematic for this kind of concentration data if its got a really skewed distribution. I would suggest you should probably also include a median and maybe an upper limit (or 90th percentile).

L156: Given its stored hive products and that they are going to be eating them why are you using contact LD50? Seems an odd decision.

L158: what are the unit for the LOQ. How are you treating values below the LOQ but > LOD, and likewise less than the LOD.

L160.5 (the table) Is this risk? Isn't this a hazard quotient as its concentration in pollen (ug/kg) / LD50 (ug bee). See the Thompson paper I reference above, this is still valid for ranking active ingredient hazard, so it works for what you are doing. But as you don't know exposure relative to

your measure of toxicity your not capturing risk I think.

Fig 1 = You see my point about effect size differences. Theres a lot of scatter here, and I appreciate that its field ecology data and that your random effects of sites probably account for some of this variability, but getting an idea of effect size differences say between the lowest 25% HQ hives and highest 25% HQ hives would bolster your arguments significantly.

Methods

L306: IS that enough colonies given inter colony variability - I know the EFSA documentation 2013 has a power analysis that considers this issue in relation to honeybees. It think on a practical level you have found effects, so you did have the power to do this. I guess theres an argument to say that if you have more colonies within sites, you would have reduced variability and your prediction and associated variance around means would go down. Either way it's a lot of colonies and a study of massive scope - I appreciate resources are not infinite.

L311: So are you focusing on average colony weight change in your analysis, or are you treating each hive as its own replicate and specifying a hierarchical random effect to account for this. Would you expect colonies with a lighter starting weight to be disproportionately more sensitive to pesticide exposure - did you consider this as a covariate (I guess ill find out). Possibly worth pointing out that average hive starting weight was not correlated with reported metrics such as hive weigher change or indeed your summed HQ values.

L326: for me this remains the big failing of this study - I appreciate you cant to everything but this is such a big consideration that you seem to have missed out. You acknowledge the importance of queen and male reproduction repeatedly and then opt not to measure it properly. You could have made a strong argument that you may have been better off harvesting one colony for residues at peak exposure and leaving others for an assessment of queen reproduction that is robust. I think this hole in the study - along with more explanation of effect size differences needs to be considered in the discussion of the main paper as a caveat..

L349: cover this summary in main text - its impressive - also scope how many of them are insecticides, fungicides and herbicides, and what proportion are currently approved.

L366: No issues with this, but see numerous earlier points about how this is not strictly risk. It's a hazard quotient.

L368: So big question here (because the table suggests you are just using honeybees) but whats this LD50 for. Is it honeybees, or is it bumblebees where you can find it (which I expect is for a handful of your 277 compounds) + an EFSA style 1/10 of the Apis LD50 (which is very protective and I doubt reflects actual toxicity in many cases). Given you are using hazard I suspect using the honeybee LD50 is as good as any and at least standardised across your compounds. Could be clearer though.

L373: Why? Pollen is eaten - they don't spray it on each other. You should use oral LD50 - I could make an exception for a product with no oral LD50 (I know there are some), but in general I don't understand why you are averaging two different toxicity endpoints.

L357: I am not saying they are not exposed to contact but everything you measure isn't really resting that. Its oral exposure at the main mode of toxicity for a stored hive product. Trying to infer contact exposure from contaminated pollen on pollen baskets seems unlikley to me.

L367: I get why you did this - to be honest I think as mentioned above it is a better approach than using some bombus toxicity data and using 1/10 Apis LD50 as a correction for the rest - it would mess with your results as some products would be disproportionately toxic.

L381-384: Could you express this as a formula rather than written.

L379: Again this is not a measure or risk - consider the ecotoxicology terminology. Also whats the multiplied by site detection frequency - presumably given you have 3 colonies a site that .33, .66 or 1. What's this actually doing as you are only measuring residues in I think 0.5 g of extracted pollen. Surely there's a lot of scope anyway for missing residues in a colony given this.

L389: why are some models with transformed response, and others you are using an error distribution like binomial.

L391: Was spatial autocorrelation an issue in the model, and if it was how did you account for it.

Referee #3 (Remarks to the Author):

This seems like an interesting, timely and significant study that substantially moves along our understanding of how real-world pesticide exposure via pollen collection might be affecting the health and potential reproductive success of a common bumblebee species (*Bombus terrestris*) in Europe. While environmental risk assessments of pesticides for insect pollinators really only consider the potential impacts of exposure to one (or perhaps a few if co-formulated in a single product) active ingredients for honey bees, the reality for honey bees and non-*Apis* bees is that they encounter the residues of multiple pesticides in the field, particularly in areas of mixed agricultural cropping systems. In this paper the authors set out to place managed bumblebee colonies in landscapes containing either apple or oilseed rape crops that also varied in the extent of crop lands (within a 1 km radius of each experimental field site). While previous studies have used such a sentinel colony approach to quantify pesticide residues collected by bumblebees (and honeybees), this study is both more highly replicated and aims to compare the risks associated with combined pesticide exposures with colony growth and productivity at a landscape level.

I have some queries and concerns about the study and the manuscript that I would like the authors to consider going forward:

This experiment was clearly a major logistical and financial undertaking as they put out 384 colonies at 128 agricultural sites (3 colonies/ site) in eight European countries. While some colonies ($n = 5$) were lost from the experimental design due to unavoidable incidents (e.g., farm machinery or wildlife related damage), quite a large number of colonies ($n = 63/384$ – i.e., 16.4%) were not included in wider analyses due to “insufficient collection of pollen” for pesticide residue sampling. I would like to see a wider consideration of why these triplets of colonies at these 21 sites were unable to collect a sufficient amount of pollen for pesticide analysis while in the field – for example, how might this have been related to landscape context/ pesticide loading? I wondered whether the authors saw any relationship(s) between landscape quality and pollen collection (overall)? I would also suggest that excluding so many colonies/ sites due to this limited extent of pollen collection raises the question about appropriateness of the decision to place three colonies per site. Some discussion of whether placing more colonies per site and balancing this against sampling few sites overall, might have resulted in the loss of fewer sites from the design. Perhaps considering how many of the colonies at the 21 excluded sites would have been needed to reach the minimum quantity of pollen needed for pesticide analyses could be considered in lieu of a sensitivity analysis?

L323-327: The authors indicate that new reproductive (gynes and new males) could have left the sentinel colonies during their time in the field. This seems surprising for gynes as I thought it was standard practice for commercial *B. terrestris* colony boxes to come fitted with a queen excluder to prevent new gynes from being able to exit the colonies. If this is not/ no longer the case, it would have been simple enough to restrict the nest entrance diameter of all colonies in the field to prevent gynes leaving the colonies – thereby giving a better indication of new queen (gyne) production on a colony/ site basis. The number and sizes of gynes produced are highly likely to be one of the significant population bottlenecks for bumblebees in their annual life-cycle.

L376-377: “We used the LD50s for adult *A. mellifera* because there is incomplete toxicity data for other bee species, and where there are data, inter-taxa correlation is high^{18,19}.” The lack of lethal endpoint toxicity information for non-*Apis* bee species is clearly an issue to be addressed in our field of research – it would be much better to have been able to use contact and oral LD50 measurements for *Bombus terrestris* in these pesticide risk calculations. I am a little concerned that there isn’t a wider discussion (either here or in the supplemental information) about the potential variability in toxicity for the same pesticides across different bee taxa, and the potential limitations of only using toxicity data for *Apis mellifera* for such risk calculations. For example, Arena & Sgolastra’s (2014) meta-analysis showed a range of sensitivity in bee taxa (R from 0.001 to 2085.7) suggesting that in some cases honeybees are 1000x more sensitive to a pesticide than the other bee species, or in other cases that the other bee species was more than 2000x time

more sensitive than a honey bee. Clearly the relative toxicity varies among bee taxa and active ingredients/ pesticide groups.

Figure 1c: comparing the fitted lines in this figure with the results, I believe that the green fitted line represents a significant negative relationship (for apple crops) – however, the yellow fitted line (for OSR) is basically a flat line. Is the latter a statistically significant relationship? If not, this needs to be more explicitly specified in the legend (and possibly indicated graphically with a dashed fitted line rather than a solid one).

References

Arena, M. and F. Sgolastra (2014). A meta-analysis comparing the sensitivity of bees to pesticides. *Ecotoxicology* 23: 324-334.

Author Rebuttals to Initial Comments:

Referee #1 (Remarks to the Author):

This manuscript describes a substantial and well-designed field experiment across 8 countries of the EU examining pesticide exposure of bumblebee colonies near apple and oilseed rape crops, and the associated impact on colony growth and production during crop bloom. The results clearly show that colonies are exposed to a vast array of pesticides, and there is a risk of this multiple exposure leading to harm at the colony level, and possibly at levels which fail to meet European Specific Protection Goals. The authors present this as strong evidence for post-approval monitoring of exposure and effects of pesticides.

Originality and significance:

The manuscript is novel and highly significant, and of interest across a wide field of researchers. It will also draw the attention of policy-makers at the forefront of crop and environmental protection.

Thank you. We also believe that our work is timely, particularly in light of recent and ongoing developments in pollinator protection efforts focused on pesticide harm reduction.

Data & methodology:

The methods are robust and the data are technically sound; although it requires further explanation as to why the colonies were terminated when crop bloom finished and how the variability across countries and sites in this experimental time-frame could affect the results [see below in Suggested Improvements].

Thank you. We address this issue in more detail below.

Statistics:

The statistical approach is thorough and the additional reporting summary is clear and in order.

We appreciate that you additionally reviewed the reporting summary. Our thanks.

Conclusions:

This was a large and well replicated experiment and the results are clear so I consider the conclusions robust and valid. I think the reflection on SPGs is useful; and the sensitivity analysis gives a wider perspective on the scale of testing required for effects to be detected.

Thank you. We encourage you to revisit our SPG analysis, which we believe is greatly improved.

Suggested improvements:

1) The intervals between colony weight measurements were based on crop bloom and varied a lot between sites (Fig S1c) – it looks like difference between g-initial and g-max varied between 10 days (Estonia, OSR) and 40 days. The authors do explain the caution required with interpretation (L106-115) but it would be helpful to do an analysis to test whether the experimental duration affected the results. Could ‘days’ be added as a co-variate? Or can we at least have a plot of days between g-initial and g-max versus total production?

This is an astute point, thank you for raising it. First, we would like to clarify that days between g-initial and g-max is not equivalent to total days of the experiment. Colonies were terminated after the third weight. In most cases (62%) the 3rd weights were the max weights, but in some cases (26%) maximum weight was achieved at the second weighing, and in other cases (12%) colonies never

gained weight. Therefore, considering your request, we have produced a plot of total days of the experiment versus total production:

While we believe the above shows the influence of experimental duration on colony production, we think your general concern is valid. We tested adding total days as a covariate to our main models (e.g., weight change model), and find that the results remain qualitatively unchanged (i.e., the interactive effect of proportion agriculture and risk is significant; $X^2 = 7.60$, $P = 0.0058$). However, we are hesitant to add this covariate because we believe this option is not robust to the data collected for the following reasons:

1. It would assume linear growth of colonies, when in fact this follows an exponential growth and decline (Duchateau & Velthuis 1988), which unfortunately we cannot analyse because...
2. We do not have fine scale data (i.e., time series of weight change or production) on colony dynamics. Ideally, one would use a colony growth model to analyse colony weight development over time (e.g., Rundlöf et al. 2015) or to create individual growth curves (e.g., Crone & Williams 2016), the parameters of which could then be analysed as we have done here.

We had to balance weighing frequency with multiple coinciding project demands (Hodge et al. 2022). There is room for future work that collects fine-scale weight data (e.g., through automation), to support examination of longer-term reproductive output and weight dynamics in relation to landscape exposure and risk. We believe our caveat on L126-130 holds, but we have modified it slightly in light of the points above:

“However, our approach meant colonies were at sites for different durations (apple: 36.3 ± 11.4 days (mean \pm SD); oilseed rape: 43.0 ± 12.2 days; Fig. S5) depending on region- and crop-specific bloom periods, precluding examination of full reproductive output and weight dynamics over the complete colony cycle, which follows an exponential growth and decline.”

Duchateau, M. J., & Velthuis, H. H. W. (1988). Development and reproductive strategies in *Bombus terrestris* colonies. *Behaviour*, 107(3-4), 186-207.

Rundlöf, M., Andersson, G. K., Bommarco, R., Fries, I., Hederström, V., Herbertsson, L., ... & Smith, H. G. (2015). Seed coating with a neonicotinoid insecticide negatively affects wild bees. *Nature*, 521(7550), 77-80.

Crone, E. E., & Williams, N. M. (2016). Bumble bee colony dynamics: quantifying the importance of land use and floral resources for colony growth and queen production. *Ecology letters*, 19(4), 460-468.

2) In fact, for Fig S1c - why have you plotted days between 1st & 2nd weights and between 2nd and 3rd weights when you only analyse the weight change between the 1st and 3rd weights? It would be clearer to be able to see the number of days between 3rd (g-max) and 1st (g-initial) weights.

We wanted to be transparent with all measurement intervals. In most cases (62%) 3rd weights were the max weights, but in some cases (26%) maximum weight was achieved at the second weighing, and in other cases (12%) colonies never gained weight (hence the 0 weight change values). We now report these proportions in the methods (L346).

3) As colony termination was also timed with crop bloom rather than colony dynamics, then the measurement of colony production is only relevant to that period and not to the whole colony cycle. Whilst you have explained caution of interpretation, it would again be useful to have a numeric demonstration of whether this experimental duration affects the actual results - in this case a plot of days between g-initial and g-max versus total production would be useful.

Please see response to your comment #1, and changes made in light of your comment #7.

4. The writing style is a bit 'sloppy' in places which makes the reader have to read sentences twice before getting the exact meaning. Please go through and clarify as much as possible.

We have endeavoured to tighten up the writing. Please see the track changes file to see complete documentation of all changes.

5. The colour-shading of the different countries in the figures is not very friendly to colour-blindness with lots of greenish and reddish tones. Can this be improved?

Good catch. We checked our figures in the COBLIS colour blindness simulator and agree the country colours are not colour-blind friendly. For country colours, we have instead opted for a Color Universal Design palette (Okabe & Ito 2008). We have updated all manuscript figures using this palette. Our plots comparing crops (i.e., with green and yellow) did not appear to be indistinguishable via the COBLIS colour blindness simulator, and are thus unchanged.

Okabe, M., & Ito, K. (2008). Color Universal Design (CUD) – How to make figures and presentations that are friendly to colorblind people. from <http://jfly.iam.u-tokyo.ac.jp/color/>

Minor revisions:

6. L57-58: this sentence needs unpacking a bit.

Agreed. Referee #2 thought so as well. Please see our response to their comment #7. We have changed this sentence to (L60):

“As central place foragers, bees' fitness depends on the net value of forage resources within their foraging range, which can be reduced if these resources are contaminated with hazardous pesticides”

7. L119: Based on the experimental time windows it is more appropriate to describe bumblebee colony performance as “colony performance during crop bloom” or something to denote that it doesn't relate to the whole season.

Yes agreed. We have made this change throughout the manuscript.

8. Fig 1a: The scale doesn't allow reader to see relative location of apple and osr points for each country. Could you put 8 insets into the supplementary material showing distribution of these sites within each country more clearly.

We cannot depict the exact location of sites given farmer confidentiality. However, we have added a supplemental 8 panel figure (Fig. S5a) that shows the points with a spatial jitter, so that their relative positions are preserved.

9. Table 2: Title needs to say 'Ten compounds *found in the colony pollen stores* that pose most risk.....'. Otherwise it sounds like you might have got the data from somewhere else.

Good catch. We have added this here and to Tables S1, S2

10. L527: Fig S1 legend: State that the x axis in plots c and d relate to the country. For clarity add '(c)' before 'The interval between.....'

Thank you. We have slightly reworked this figure. We have replaced the (redundant) Europe scale map with the individual country maps. Old Fig. S1b is now an additional supplemental figure (Fig. S9a). We state that the colors in (now) b and c relate to the country colors in a.

References:

The manuscript is well referenced and set well within the context of this controversial field.

Abstract & Overall:

The abstract provides a very good summary of the work, and the conclusions are appropriate. The findings are important and of general interest.

Thank you for your careful review of our manuscript, its figures, and the reporting summary.

Referee #2 (Remarks to the Author):

This paper describes field monitoring of bumblebee colonies (>300) across numerous agricultural locations (>100) in 8 European countries. The paper reports widescale prevalence of pesticide residues within the colonies and associates this with reduced viability of those colonies. I think this is a potentially important study, but I have some concerns over the interpretation (to an extent, I think what they have done is good). I have some issue with terminology (risk vs hazard, and that they don't really measure risk per se - however given the correlative nature of this I don't see this as a major issue, they just need to be more careful in their terminology).

I think my major criticisms are:

1) You basically ignore queen production (you touch on it a bit) on the basis you sampled hives too early for this to be meaningful. This is such a major factor for understanding reproductive potential of hives and so pop. Persistence - you may show a reduction in hive weight but if this isn't having a major impact on queen / male production is this actually a risk. Certainly you could argue yes, but for me this would be a definitive argument. What you show may be a risk - that said I still think your findings are important;

Thank you raising this point. We now highlight queen production responses that were previously included in the supplement, while continuing to caveat this result. Additionally, as part of addressing your points regarding effect sizes, we have included queen production as a response here as well. Please see our more detailed responses and additional results below.

2) You need to make more use of effect sizes somehow to indicate the scale of the effect - you do at one point but I am unclear how it was derived or what it compares to. I think comparing say effect size differences between your 25% lowest HQ colonies vs 25% highest Hq colonies, or relating it to 90th percentile HQ colonies to average would be valuable - your data has a lot of scatter and I want to know if this is an important or trivial difference. This is easy to do I suspect but may require some re-adjustment of interpretation. I think if the queen data was more robust it would be more of a slam-dunk for me, but I still think given the scale and scope of the study its Nature worthy - I just want more insight from you into the results as to whether the trends are of trivial importance or actually represent large effect sizes that affect major parts of the population.

We believe these comments have greatly improved the manuscript, so our thanks for pushing us to include effect sizes. We have performed a new series of analyses related to our original Specific Protection Goal (SPG) figure. These analyses include a refinement of the SPG figure and comparisons of effect between risk groups, as you requested. Please see our more detailed responses and additional results below.

Abstract

1. L37-39: not sure why you need to make such a big point about neonics here unless they are explicitly a major risk (which given the ban in the EU outside of some derogations seems less likely, but not impossible). I think its fair to say that risks to bees from insecticides (as well as other synergistic effects) are pretty likely (thus the regulatory process) so I don't see the need for a line saying neonics ore bad so other insecticides may also be bad.

With this sentence we make the point that besides neonicotinoids there is no field-base evidence of the effects of landscape exposure. Perhaps this point has become lost, so we have revised this sentence (L36):

“Field studies demonstrated that agricultural use of neonicotinoid insecticides can negatively affect wild bee species^{1,2}, leading to restrictions on these compounds³. However, besides neonicotinoids, field-based evidence of the effects of landscape pesticide exposure on wild bees is lacking. Bees encounter multiple pesticides in agricultural landscapes...”

2. L40: What do you mean by growth and development? Are you talking about individuals or a colony. I would argue that a lot of scientific studies (neonics to quote your example) impacts on growth and development of honeybees, bumblebees and Red Mason bees have been widely tested. Higher level Risk assessments also look at colony growth effects (see Pilling, E., et al (2013) A Four-Year Field Program Investigating Long-Term Effects of Repeated Exposure of Honey Bee Colonies to Flowering Crops Treated with Thiamethoxam. PLoS ONE, 8, e77193) and arguably lower tier work looking at chronic toxicity of honeybees and increasingly other species (although that's still limited) look at impacts on a range of lifestages from larvae to different casts. I kind of get your point but I think your overplaying this here - I think its the scale and consistency of your assessment that makes this exciting. Also you keep saying bees - what bee species are you talking about.

Thank you for asking us to clarify. To your points:

- 1. Growth and development of the reproductive unit, so colony or nest. We have added 'colony' to this sentence to clarify*
- 2. We do not think this point is overplayed. There are two important gaps that this study addresses: 1) mixture effects outside the lab and 2) agricultural use driven exposure beyond neonic semi-field experiments (one substance, one or a group of adjacent fields).*
- 3. Our point applies to all bee species. We have added 'species' to this sentence to clarify*

We have revised the summary and first paragraph of the intro to make this point clearer.

3. L42: I am finding the work associated a bit on the vague side. Is this that you have experimentally demonstrated reduced colony performance from exposure, or you have a list of pesticides and you are inferring potential impacts from a risk quotient or something like that, or your just saying there are pesticides in the hives and they might have an impact (I am guessing not the latter). Needs to be more specific.

None of these options. We say 'associated' because we correlated colony performance with risk through regression analysis. We believe that this sentence is clear within the context of the manuscript and analysis. This risk results from landscape exposure, which is the outcome of multiple pesticide uses (each with potentially different environmental fate and toxicities) intersecting with bee activity. We expand on this in more detail in our response to your comment #16.

Main text

4. L52: So that's also not true - there is a massive body of work looking at mixture effects and toxicity, from concentration addition models, TKTD models, NUTS models etc. Synergisms are defined as where they deviate from model predictions about additive toxicity of similar or different modes of action - see Cedergreen, N. (2014) Quantifying Synergy: A Systematic Review of Mixture Toxicity Studies within Environmental Toxicology. PLoS ONE, 9, e96580 for an example of this. All these models are intended to predict field level effects. Again I get what you are aiming for, that large scale experimental testing of mixture effects hasn't been done, but I also think your kind of ignoring the massive areas of ecotoxicology that does look at this to drive home your point.

Specifically if you are looking at trying to show field level synergisms that these need to be relative to mortality effects you would expect from a some kind of mixture toxicity model - again see the cedergreen paper.

There is indeed a body of literature focusing on mixture toxicity and effects in bees, but these studies are done in the lab or under semi-field conditions and often for binary mixtures (as concluded in the review by Tosi et al. 2022). The gap that we are highlighting here is not this mechanistic understanding of mixture toxicity/effects but rather the effects of landscape pesticide exposure – the related risk, and the consequences for bees in European agricultural landscapes. Tackling such realistic co-exposure scenarios is a remaining challenge in general in chemical risk assessment given the number of substances, limited toxicity information (beyond single substances, for lethality and specific model organism at individual level) and the infinite number of combinations (Bopp et al. 2019). Here we focus on exploring the bee consequences of landscape exposure using additive toxicity weighted concentration, we are satisfied with the assumption of risk additivity of the pesticide mixture. Using concentration addition has repeatedly been highlighted as a reasonable, even conservative, assumption for handling chemical mixtures (e.g. Cedergreen 2014, Bopp et al. 2018, EFSA Scientific Committee et al. 2019) and it is even the main conclusion in Cedergreen (2014) to use the CA or similar approaches in chemical mixture ERA. In addition, we have now worked to further clarify ‘landscape exposure’ (L54, Fig S1), which is the reality for bees in agricultural landscapes, and the resulting risk when accounting for both exposure and hazard (please see our response to your comment #16).

Bopp, S. K., Barouki, R., Brack, W., Dalla Costa, S., Dorne, J. L. C., Drakvik, P. E., ... & Bergman, Å. 2018. Current EU research activities on combined exposure to multiple chemicals. Environment International 120: 544-562.

Bopp, S. K., Kienzler, A., Richarz, A. N., van der Linden, S. C., Paini, A., Parissis, N., & Worth, A. P. 2019. Regulatory assessment and risk management of chemical mixtures: challenges and ways forward. Critical Reviews in Toxicology 49: 174-189.

Cedergreen N 2014. Quantifying Synergy: A Systematic Review of Mixture Toxicity Studies within Environmental Toxicology. PLoS ONE 9(5): e96580.

EFSA Scientific Committee & More, S.J. et al. 2019. Guidance on harmonised methodologies for human health, animal health and ecological risk assessment of combined exposure to multiple chemicals. EFSA J. 17, e05634 (2019).

Tosi, S., Sfeir, C., Carnesecchi, E., & Chauzat, M. P. 2022. Lethal, sublethal, and combined effects of pesticides on bees: A meta-analysis and new risk assessment tools. Science of The Total Environment 844: 156857.

5. L53-54: I don't think this statement is true - see points above. I can see what you are getting but there are multiple studies that look at this, including ones by authors on this paper.

Please see our response to you comment #4

6. L54-56: Although I think you need to be careful about the language and some of your statements, I do think this is important and its an impressive body of sites / colonies that underpins your assessment.

Thank you.

7. L58: not sure I like the ‘nutritional benefit reduced by pesticide cost’, again I kind of see what your saying in terms of a colony scale effect, but I think its a weird way of saying it. I also think its more likely to be relevant for chronic as opposed to acute toxicity effects and relative to the scale of exposure. Acute toxicity for a solitary species makes the effects of forage quality irrelevant. Also what about all the other drivers of population change like predation, disease, nesting site availability, landscape structure etc. Seems a bit of a simplified statement. That said statement in L58-60 seems reasonable as a working hypothesis.

Good point and we agree. Other factors, such as those you highlight, could also reduce the nest or colony-level benefit of foraging. The syntax of this sentence made it seem like pesticides are the only factor that could reduce this net value. We have clarified this sentence (L60):

“As central place foragers, bees' fitness depends on the net value of forage resources within their foraging range, which can be reduced if these resources are contaminated with hazardous pesticides^{7,8,17}.”

8. L63-65: Sounds great - a really impressive sampling network.

Thank you.

9. L65: Can you do an overview of what active ingredients you are looking at. How many, are they just insecticides, or are there fungicides and herbicides. Does this cover both current licenced products or do you include out of licence products as well.

Good idea. The 267 compounds screened for include fungicides, herbicides, and insecticides. These 267 compounds include compounds both approved and not approved under Reg. (EC) No 1107/2009. We have added a short summary sentence (L70):

“We collected pollen samples from the colonies, which were screened for 267 compounds (Table S1) to quantify pesticide residues”

We have added a supplemental table (Table S1) that includes the 267 screened compounds with their EU registration status, pesticide class (e.g., fungicide, insecticide), and reference to the protocol paper for this compound screening (Kiljanek et al. 2021).

Kiljanek, T., Niewiadowska, A., Małysiak, M., & Posyniak, A. (2021). Miniaturized multiresidue method for determination of 267 pesticides, their metabolites and polychlorinated biphenyls in low mass beebread samples by liquid and gas chromatography coupled with tandem mass spectrometry. *Talanta*, 235, 122721.

10. L66-71: this is a really important finding. When you say cropland moderated this effect do you mean this is because there is less crop (so less pesticides) or is this due to greater food resources from semi-natural. This could do with more explanation. Is this having an effect as an interaction, or is it just another covariate in the model. Its not clear from reading the text.

Thank you and good point. We could have been more clear that this is an interaction effect, although Table 1 does corroborate this. To make this clearer, we have revised the text as (L76):

“We found that increasing pesticide risk... reduced bumble bee colony production, and this effect was moderated by an interaction with the proportion of cropland in the surrounding landscape.”

Our interpretation of this finding is given on L94:

“Simplified landscapes, dominated by non-flowering cropland, generally contain fewer flower resources^{23,24}, potentially stressing colonies and interacting with pesticide effects^{25,26}. Likewise, high pesticide risk may hamper the bees' foraging efficiency²⁷, an already difficult task in resource-poor environments.

11. L69: so I guess one question I have is whether you had a look at Queen production. Given you could argue these are the reproductive units (see Rundlöf, et al (2015) Seed coating with a neonicotinoid insecticide negatively affects wild bees. *Nature*, 521, 77-80; Whitehorn, P.R., et al (2012) Neonicotinoid pesticide reduces Bumble Bee colony growth and queen production. *Science*,

336, 351-352) this seems an important thing to consider. At the end of the day if queen production is not negatively effected this is an important finding, even if colonies on average have lower weight gain or total number of cells. Without this you do have an important finding that certainly suggests there may be effects, but from a risk management perspective you could argue that if it doesn't impact on population viability (presumed from say queen production) is it a problem - probably yes it is although more subtle, and certainly you could argue it would be important for solitary species and for ecosystems service provision (where numbers of workers do matter), or for other bumblebee species that tend to have smaller total worker numbers to start with. I think you need queen production trends or if not an explanation that this may be important.

We are aware of the importance of measuring queen production. This comment surprised us because we included queen production trends in the original submission (old Fig. S3), along with a caveats for why it is not included as a main finding:

“Indeed, we see evidence that the production of new queens declined with increasing risk similarly to weight change (Table S1, Fig. S3). However, our approach meant colonies were at sites for different durations (apple: 36.3 ± 11.4 days (mean \pm SD); oilseed rape: 43.0 ± 12.2 days; Fig. S1c) depending on region- and crop-specific bloom periods, precluding careful examination of longer-term reproductive output, which follows an exponential growth and decline of the colony.”

Despite these caveats, we do expect an impact on queen production because

1. *Our results show an effect (Fig. S4, see also our response to your comment #24) and*
2. *Queen production and colony size (e.g., weight) are correlated (Müller & Schmid-Hempel 1992, Westphal et al. 2009).*

We have provided additional figures and analyses that show the correlation between our response variables (Fig. S3):

*Müller, C. B., & Schmid-Hempel, P. (1992). Correlates of reproductive success among field colonies of *Bombus lucorum*: the importance of growth and parasites. *Ecological Entomology*, 17(4), 343-353.*

*Westphal, C., Steffan-Dewenter, I., & Tschardtke, T. (2009). Mass flowering oilseed rape improves early colony growth but not sexual reproduction of bumblebees. *Journal of Applied Ecology*, 46(1), 187-193.*

12. L66: Also general question is I think it would be useful to given net effect size changes in your key metrics over potentially: 1) the range of risk (or hazard quotients) you are finding in the hives; and 2) at a 90 percentile HQ values relative to the lowest HQ value to link in (kind of) with EFSA documentation on protecting 90% of the population / 90% exposure risk. This seems to be the protective standard they are going to so you may as well make use of this.

Good point. We address this in our responses to your comments 24, 25, & 26

13. L73: So you do this repeatedly, you say cropland or crop moderated an effect - can you say how it moderated it. I appreciate you have limited space and a lot of results - but if it's a consistent overall trend in the way cropland had an effect maybe just describe this so its clear, but note there are exceptions.

We subsequently discuss the details of the interactions (L84, L88). As you note, space is limited. We tried several reporting orders, and this style (main effects followed by description of marginal effects) struck the balance between concision, clarity, and correspondence with graphical elements.

14. L75: What do you mean by mor resource yielding - I am guessing you mean in terms of per area pollen / nectar. If this isn't something you tested (which I think it isn't as you have a reference) maybe save this statement for the discussion as it seems slightly out of place here.

Your interpretation is correct (and is what the study by Baude and colleagues (2016) provides). We believe it is important to make this evidenced point because it substantiates the inference in the second clause of this sentence.

*Baude, M., Kunin, W. E., Boatman, N. D., Conyers, S., Davies, N., Gillespie, M. A., ... & Memmott, J. (2016). Historical nectar assessment reveals the fall and rise of floral resources in Britain. *Nature*, 530(7588), 85-88.*

15. L77: Ok you seem to have these overall summaries I asked for (I think) in the moderating effects of crop land. I would put these earlier and then give the details after. Or certainly this is a model for how to describe each moderating effect.

Thank you. Please see our response to you comment #13.

16. L81: Is it risk or hazard you are measuring. I.e. is it exposure (ng. bee-1) / toxicity (e.g. LD50 ng bee-1) or a measure of hazard where you don't have exposure per bee, but have a measure of residues concentrations. Its ecotoxicology what you are doing so I think you need to be careful about terminology.

We take a general, ecological perspective on bees' pesticide-related risk, quantified as additive toxicity-weighted concentrations. This metric allows us to explore relative differences in bees' pesticide exposure (concentration and identity) across the landscape, disentangled from estimated variable consumption rates to colony size. To our mind, exposure - the degree to which an organism encounters pesticides at a given time and place (Sponsler et al. 2019), does not need to account for consumption. However, consumption is required to calculate a dose (Gradish et al., 2019). Our distinction between exposure and dose also follows that of the US EPA. Furthermore, we use this pesticide risk metric in our previous work exploring pesticide exposure (Knapp et al., 2023) and effects (Rundlöf et al., 2022) between bee species and landscape contexts – indicating the relevance and applicability of this metric. We now illustrate and clarify these points and our approach through a new supplemental figure (Fig. S1):

Figure S1 A simplified view of landscape exposure and resulting pesticide risk to bees. Pesticide use creates potential hazard for non-target organisms. For bees in agricultural landscapes, pesticide risk results when their activity exposes them to this hazard (top left panel). Without the co-occurrence of hazard and exposure we expect no risk (remaining panels). Of course, the degree of hazard and exposure will depend on pesticide properties (e.g., toxicity, environmental fate, product formulations, use patterns) and bee traits (e.g., foraging range, sociality, body size, detoxification pathways). Moreover, real-world exposure occurs at landscape scales (see insets), because bees can integrate multiple sources of exposure by visiting spatially separated patches that vary in the identity, amount, timing, and toxicity of hazard. We use colony pollen stores to quantify pesticide risk resulting from this landscape exposure. We quantify exposure as the concentrations (ug/kg) of 267 substances in the pollen while hazard is quantified by the substances' toxicities (LD50s). Scaling concentrations by toxicities and summing these toxicity weighted concentrations provides a relative measure of pesticide risk to bees.

Sponsler, D. B., Grozinger, C. M., Hitaj, C., Rundlöf, M., Botías, C., Code, A., ... & Douglas, M. R. (2019). Pesticides and pollinators: A socioecological synthesis. *Science of the Total Environment*, 662, 1012-1027.

Gradish, A. E., Van Der Steen, J., Scott-Dupree, C. D., Cabrera, A. R., Cutler, G. C., Goulson, D., ... & Thompson, H. (2019). Comparison of pesticide exposure in honey bees (*Hymenoptera: Apidae*) and bumble bees (*Hymenoptera: Apidae*): implications for risk assessments. *Environmental Entomology*, 48(1), 12-21.

Rundlöf, M., Stuligross, C., Lindh, A., Malfi, R. L., Burns, K., Mola, J. M., ... & Williams, N. M. (2022). Flower plantings support wild bee reproduction and may also mitigate pesticide exposure effects. *Journal of Applied Ecology*, 59(8), 2117-2127.

Knapp, J. L., Nicholson, C. C., Jonsson, O., de Miranda, J. R., & Rundlöf, M. (2023). Ecological traits interact with landscape context to determine bees' pesticide risk. *Nature Ecology & Evolution*, 7(4), 547-556.

17. L83: See earlier point about I want to know the range and types of pesticides you are assessing - just a few lines earlier would be good.

Thank you. Please see our response to your comment #9.

18. L86: Again is this a risk, or is this hazard. If you are dividing residues in pollen / toxicity you are not using the same units so its not really a risk, its just a scaled measure of hazard. Nothing wrong with that in the context of this study as its correlative looking at relative toxicity, but just be careful about language. Note I appreciate some HQ have been quantified to give a measure or risk (e.g. application to crop / LD50 has defined levels of concern, but those were experimentally validated.

See Thompson, H.M. (2021) The use of the Hazard Quotient approach to assess the potential risk to honeybees (*Apis mellifera*) posed by pesticide residues detected in bee-relevant matrices is not appropriate. *Pest Management Science*, 77, 3934-3941. For a more detailed discussion of some of these issues).

Please see our response to your comment #16.

19. L87: Most insecticides are bee health antagonists - they kill them with sufficient exposure. Are you talking about pesticides that may have synergistic interactions that results in a deviation in toxicity from predicted levels, or specifically impacts of some measure of bee health at an individual or colony level. Again I get what you are aiming for but the language needs tightening up.

Good point and we agree that most insecticides are bee health antagonists. We wanted to call out these compounds because there is a substantial evidence base of their individual effects (hence the use of 'known'). We think that as a general statement this term is fine.

20. L89: that needs more explanation in the main text. I am not sure what you are getting at here. I need to look at the methods maybe to get a better idea, but this seems a weird metric - also I have no idea what the range of 1- 3.8 relates to in terms of a Level of Concern (e.g. like a RQ > 1 for acute toxicity). Is it good or bad, or a unitless scale that just goes from presumed less bad to bad.

We believe that this metric (maximum cumulative ratio, MCR) is adequately described in the clause and thoroughly described in the methods. However, we have substituted the term 'risk ratio' for the published name of this metric. MCR is simply a unitless dominance measure that indicates the relative contribution to mixture risk from the single most risky compound. As we state (L413):

"When MCR = 1, risk comes from a single compound; thus, the MCR represents the factor by which the pesticide mixture is riskier than the single most risky compound."

For the equation of this metric, see also our response to your comment #43

21. L99: Or just the widespread use of azole fungicides for arable farming could do the same.

Good point. We have changed the 'and' to 'or' to make this point.

22. L101: Agreed, however it could also overestimate it if there are antagonistic effects. Again see the Cedergreen paper. However, I am in general agreement with you that mixture effects risks are potentially hard to estimate. That said you could potentially test for this as you know where certain common synergistic products are found - e.g. azoles. You could include them as a covariate in your model as part of an interaction to see if where present you get greater than expected negative effects on your colony metrics. However, just a thought - I can see it's a bit of a throwaway line and not an unreasonably one so not asking you to reanalyse your data.

Good idea, but beyond the scope of this manuscript. Please see also our response to your comments #4 and 23

23. L102-103: Can you expand on this - its one of the its that sounding interesting but its not entirely clear what you are getting at.

This sentence was actually based on the Cedergreen 2014 paper that you point to in comment #4. Since exploring synergism is not within the scope of our manuscript and building on the conclusions in Cedergreen (2014), we have replaced this sentence with (L113):

“Nonetheless, synergism among pesticides is relatively rare (Cedergreen 2014) and assuming concentration addition is considered a reasonable starting point in regulatory risk assessment of mixtures (EFSA Scientific Committee et al. 2019).”

24. L108-109: I think you need to have the males and queen reproduction to back up your argument as a response. You kind of touch on this in the next section but I think you need to go into more detail. Leaving it out is a major hole in your arguments. While colony weight can be correlated with queen numbers, I have seen papers where its not (I am going to be honest I cant remember which one). I think your arguemtyn holds, but it does undermine this by not having an explicit demonstration that queen production was reduced (and more importantly reduced by how much). This comes back to the effect size issue again. Ok yes you may have found a negative effect, but if from an effect size (as opposed to just comparing means and ignoring variance) perspective its 1-2% difference when comparing the best and worst case situations that’s not that worrying, and sits within accepted thresholds. I guess this is the difference between whether what you report is unacceptable risk - 90% of colonies show a reduction in colony queen reproduction of >20% for example vs acceptable risks like 90% of colonies show a reduction in queen reproduction of <2%. This needs to be better qualified in the main paper text. It might not be as bad as it could be, but it would be scaled to be meaningful.

Thank you for this comment, it pushed us to provide additional analyses that we believe greatly strengthen the manuscript (Fig. S6). Based on your suggestions, we have grouped colonies into those that belong to less than the 25th percentile and greater than the 90th percentile of risk. We then compare max weight, total production, and queen production between these risk groups:

We discuss these additional results (L148):

“Further, compared to low-risk colonies, we observed a 34% reduction in maximum weight (estimated mean difference: 393 g; Fig. S6c), 52% reduction in total production (410 individuals; Fig. S6d), and a 47% reduction in queen production (21 individuals; Fig. S6e) in the high-risk group (i.e., the 90th percentile of risk).”

25. L130 - fantastic you have this - can you do more of this. Specifically for the queen production if you can. That said I don’t get what you mean by using the average maximum colony weight across the colonies as reference. Are you saying when you compare to the worst possible case of pesticide hazard or risk based on residues there is a 10% reduction in colony size. This seems unquantified to me. What’s the difference say compared to a 90th percentile worst case scenario of hazard relative to the mean. Its good though to have these figures and relate to a protection goal.

Thank you. Considering your previous suggestions, we have re-specified the baseline condition that we reduce weight by 10% in accordance with the suggested SPG for bumblebees. Now, we take the

average maximum weight of colonies belonging to the first quartile of risk (i.e., your suggestion for “25% lowest HQ colonies”). For the remaining colonies we calculate what proportion are above and below the 10% reduction SPG:

To your comment #26, we can see that 60% of colonies are below this SPG (a), and that these colonies generally had higher risk (b, $\chi^2 = 6.85$, $P < 0.01$). Please see our responses above for comparisons between low and high risk groupings for our colony performance measures (including queen production).

We have included these, together with the comparisons described above as a new supplemental figure (Fig. S6), which we discuss (L143):

“Using the average maximum weight of low risk colonies (i.e., the 25th percentile of risk) as a baseline, we found that 60% of remaining colonies exceed a current suggested Specific Protection Goal (SPG) for bumble bees (10% colony weight reduction⁴², Fig. S6a) and that these colonies were more at risk (Fig. S6b). Further ...”

26. L132: Again this surely depends on what % of the population is reduced by 10% relative to the mean.

Please, see our comment and analyses above. We believe that together these results support our statement:

“Thus, the European pesticide regulatory system for pesticides is not sufficiently protective given this SPG, evidencing the need for post-approval monitoring of landscape exposure and its effects”.

27. L135: Is this a post hoc power analysis. I don’t get what your saying - maybe this would be better covered in the methods but acknowledge in the main text the need for lots or replication. Specifically what I don’t get with this statement is that by definition you had the power to detect the effects you saw (you have a p value to prove it) with the replication you had (which was 100 ish sites). Normally this would be to say you couldn’t detect an effect size of say 5% with the replication you had so effects at that scale were undetectable. This may just be better in the methods where you can explain it a bit more as I feel I am missing something here.

Yes, in a way this is a power analysis, but we hesitate to call it that because it is post-hoc. Our point here was not to explore the level of replication needed to detect an effect of specific magnitude. The point being that these types of studies simply require high levels of replication to detect any effect.

28. L140-145: Overall I really agree with what you are saying. Lower tier risk assessments focus on individual toxicity, not colony level effects so much, with risks here being the basis for kicking off higher tier assessments. - I guess a lot of this comes down to the failure of higher tier risk assessments that are quantifying exposure under field conditions (albeit probably more for honeybees). Although I guess on a practical level they focus on one active at a time, not field scale real world mixture exposure which is what you are quantifying. It seems to be more of a systemic failing of the regulatory approach to consider real world post regulation mixture (additive or synergistic) toxicity.

Thank you. "Real world mixture exposure" is exactly the focus here!

29. Table 2: So means are inherently problematic for this kind of concentration data if its got a really skewed distribution. I would suggest you should probably also include a median and maybe an upper limit (or 90th percentile).

Good point and we agree. We have added the median and 90th percentile concentration values to Table 2 and the supplemental table.

30. L156: Given its stored hive products and that they are going to be eating them why are you using contact LD50? Seems an odd decision.

Bees will potentially come in contact with these compounds in pollen through (1) extraction from flowers, (2) storage during transport, and (3) transfer to larvae or pollen stores. Therefore, we averaged the acute oral and contact LD₅₀ of each compound to provide an overall indicator of toxicity, reflective of how bees encounter pesticides in the landscape and their multiple exposure routes.

31. L158: what are the unit for the LOQ. How are you treating values below the LOQ but > LOD, and likewise less than the LOD.

The units are µg/kg. We now specify this in the table caption. All values < LOQ are treated as zero. We have added this point to the methods (L405).

32. L160.5 (the table) Is this risk? Isn't this a hazard quotient as its concentration in pollen (ug/kg) / LD50 (ug bee). See the Thompson paper I reference above, this is still valid for ranking active ingredient hazard, so it works for what you are doing. But as you don't know exposure relative to your measure of toxicity your not capturing risk I think.

Please see our response to your comment #16.

33. Fig 1 = You see my point about effect size differences. Theres a lot of scatter here, and I appreciate that its field ecology data and that your random effects of sites probably account for some of this variability, but getting an idea of effect size differences say between the lowest 25% HQ hives and highest 25% HQ hives would bolster your arguments significantly.

Please see our response to your comment #24 and 25. Again, this was a great suggestion. Thank you

Methods

34. L306: IS that enough colonies given inter colony variability - I know the EFSA documentation 2013 has a power analysis that considers this issue in relation to honeybees. It think on a practical level you have found effects, so you did have the power to do this. I guess theres an argument to say that if you have more colonies within sites, you would have reduced variability and your prediction

and associated variance around means would go down. Either way it's a lot of colonies and a study of massive scope - I appreciate resources are not infinite.

Indeed, balancing hive replication and site replication was one of the challenges of this experimental design. As you point out, we did detect effects, nonetheless. For practical considerations of hive-level replication please see our response to referee 3 comment #1.

35. L311: So are you focusing on average colony weight change in your analysis, or are you treating each hive as its own replicate and specifying a hierarchical random effect to account for this. Would you expect colonies with a lighter starting weight to be disproportionately more sensitive to pesticide exposure - did you consider this as a covariate (I guess I'll find out). Possibly worth pointing out that average hive starting weight was not correlated with reported metrics such as hive weight change or indeed your summed HQ values.

Exactly. We are "treating each hive as its own replicate and specifying a hierarchical random effect to account for this".

Your points about initial weight are astute. Thank you for raising them. Hive initial weight was not correlated with pesticide risk ($\chi^2 = 1.215$, $P = 0.27$). The following figure has been added as a supplemental figure (Fig S9b) and we report the above statistics in the methods (L432).

For our main models, we have now included initial weight as a covariate. Our main results remain qualitatively unchanged. We have updated all in line statistics, tables and figures to reflect this new model specification.

36. L326: for me this remains the big failing of this study – I appreciate you can't do everything but this is such a big consideration that you seem to have missed out. You acknowledge the importance of queen and male reproduction repeatedly and then opt not to measure it properly. You could have made a strong argument that you may have been better off harvesting one colony for residues at peak exposure and leaving others for an assessment of queen reproduction that is robust. I think this hole in the study - along with more explanation of effect size differences needs to be considered in the discussion of the main paper as a caveat.

The study had multiple purposes (see Hodge et al. 2022) which meant that our aims in this manuscript had to be balanced in relation to other aims. However, this did not exclude us from also including queen production as a dependent endpoint even if we present the results with appropriate caveats - please see our response to your comment #11 for details.

Hodge, S., Schweiger, O., Klein, A. M., Potts, S. G., Costa, C., Albrecht, M., ... & Stout, J. C. (2022). Design and planning of a transdisciplinary investigation into farmland pollinators: rationale, co-design, and lessons learned. *Sustainability*, 14(17), 10549.

37. L349: cover this summary in main text - its impressive - also scope how many of them are insecticides, fungicides and herbicides, and what proportion are currently approved.

Please see our response to your comment #9

38. L366: No issues with this, but see numerous earlier points about how this is not strictly risk. It's a hazard quotient.

Please see response to your comment #16.

39. L368: So big question here (because the table suggests you are just using honeybees) but what's this LD50 for. Is it honeybees, or is it bumblebees where you can find it (which I expect is for a handful of your 277 compounds) + an EFSA style 1/10 of the Apis LD50 (which is very protective and I doubt reflects actual toxicity in many cases). Given you are using hazard I suspect using the honeybee LD50 is as good as any and at least standardised across your compounds. Could be clearer though.

You are correct; we use honeybee LD50's because this is where data is most rich, and we needed a standardized measure that we could use to scale across all compounds. We now make this clearer in the legend of Table 2

40. L373: Why? Pollen is eaten - they don't spray it on each other. You should use oral LD50 - I could make an exception for a product with no oral LD50 (I know there are some), but in general I don't understand why you are averaging two different toxicity endpoints.

Please see our response to your comment #30

41. L357: I am not saying they are not exposed to contact but everything you measure isn't really resting that. Its oral exposure at the main mode of toxicity for a stored hive product. Trying to infer contact exposure from contaminated pollen on pollen baskets seems unlikely to me.

We have used the average of oral and contact toxicity because the residues in pollen stored in bee colonies is 1) the result of bees flying to different parts of the landscape during the colony life, collecting pollen and grooming it from their hairy bodies and carrying it on their hind legs (or other parts of the body in other bee species) – causing contact exposure, and 2) the food for mainly larvae and new queens – causing contact and oral exposure for the bees that feed the larvae and oral exposure to larvae and queens. Thus, we expect multiple routes of exposure for the bees and find it appropriate to use an average for oral and contact to approximate toxicity.

42. L367: I get why you did this - to be honest I think as mentioned above it is a better approach than using some bombus toxicity data and using 1/10 Apis LD50 as a correction for the rest - it would mess with your results as some products would be disproportionately toxic.

We agree.

43. L381-384: Could you express this as a formula rather than written.

Yes. Following Price and Han (2011), we now express this as (L412):

$$MCR = \frac{TWC_{mix}}{\max(TWC_i)}$$

Price, P. S., & Han, X. (2011). Maximum cumulative ratio (MCR) as a tool for assessing the value of performing a cumulative risk assessment. *International journal of environmental research and public health*, 8(6), 2212-2225.

44. L379: Again this is not a measure of risk - consider the ecotoxicology terminology. Also what's multiplied by site detection frequency - presumably given you have 3 colonies a site that .33, .66 or 1. What's this actually doing as you are only measuring residues in I think 0.5 g of extracted pollen. Surely there's a lot of scope anyway for missing residues in a colony given this.

This measure of compound-specific risk (defined in Sanchez-Bayo and Goka 2014) multiplies each toxicity-weighted concentration (TWC_i) by its site-level detection frequency across all sites, thus accounting for the exposure probability over the whole study.

For the pollen collection, the intent was to sample at least 15 g of stored pollen pooled from the three colonies at a site. However, the pooled sample was less at some sites and samples down to 0.52 g were used for pesticide residue analysis (as specified in L356). The methods also specify that the pooled sample of stored pollen from a site was homogenized before subsampled for palynology and pesticide residue analysis, so even if a smaller sample was analysed for pesticide residues, it represented an often much larger initial sample from the colonies at a site. We have made this clearer by adding "homogenized" in the pesticide residue analysis section (L376), similar to the statement in the palynology section.

Sanchez-Bayo, F., & Goka, K. (2014). Pesticide residues and bees—a risk assessment. *PLoS one*, 9(4), e94482.

45. L389: why are some models with transformed response, and others you are using an error distribution like binomial.

We use a negative binomial distribution via GLMMs for count data, such as total production. The weight data are continuous and thus we use log transformations and LMMs

46. L391: Was spatial autocorrelation an issue in the model, and if it was how did you account for it.

Our study system was explicitly designed to avoid spatial autocorrelation by ensuring all colonies were at least 3 km apart, which is sufficient given that the average foraging range of bumblebees is approximately 1.5 km (Kendall et al. 2022). We now clarify this on L326:

"All sites were >3 km apart to ensure the spatial independence of the bumble bee colonies, whose foraging range is generally <1.5 km"

Kendall, L. K., Mola, J. M., Portman, Z. M., Cariveau, D. P., Smith, H. G., & Bartomeus, I. (2022). The potential and realized foraging movements of bees are differentially determined by body size and sociality.

Referee #3 (Remarks to the Author):

This seems like an interesting, timely and significant study that substantially moves along our understanding of how real-world pesticide exposure via pollen collection might be affecting the health and potential reproductive success of a common bumblebee species (*Bombus terrestris*) in Europe. While environmental risk assessments of pesticides for insect pollinators really only consider the potential impacts of exposure to one (or perhaps a few if co-formulated in a single product) active ingredients for honey bees, the reality for honey bees and non-*Apis* bees is that encounter the residues of multiple pesticides in the field, particularly in areas of mixed agricultural cropping systems. In this paper the authors set out to place managed bumblebee colonies in landscapes containing either apple or oilseed rape crops that also varied in the extent of crop lands (within a 1 km radius of each experimental field site). While previous studies have used such a sentinel colony approach to quantify pesticide residues collected by bumblebees (and honeybees), this study is both more highly replicated and aims to compare the risks associated with combined pesticide exposures with colony growth and productivity at a landscape level.

Thank you for this insightful overview of our manuscript, you have touched on a number of topical points that we hoped this work contributes to.

I have some queries and concerns about the study and the manuscript that I would like the authors to consider going forward:

1. This experiment was clearly a major logistical and financial undertaking as they put out 384 colonies at 128 agricultural sites (3 colonies/ site) in eight European countries. While some colonies ($n = 5$) were lost from the experimental design due to unavoidable incidents (e.g., farm machinery or wildlife related damage), quite a large number of colonies ($n = 63/384$ – i.e., 16.4%) were not included in wider analyses due to “insufficient collection of pollen” for pesticide residue sampling. I would like to see a wider consideration of why these triplets of colonies at these 21 sites were unable to collection a sufficient amount of pollen for pesticide analysis while in the field – for example, how might this have been related to landscape context/ pesticide loading? I wondered whether the authors saw any relationship(s) between landscape quality and pollen collection (overall)? I would also suggest that excluding so many colonies/ sites due to this limited extent of pollen collection raises the question about appropriateness of the decision to place three colonies per site. Some discussion of whether placing more colonies per site and balancing this against sampling few sites overall, might have resulted in the loss of fewer sites from the design. Perhaps considering how many of the colonies at the 21 excluded sites would have been needed to reach the minimum quantity of pollen needed for pesticide analyses could be considered in lieu of a sensitivity analysis?

We have further investigated whether other factors could explain the inability of colony foragers to provide sufficient pollen for these sites. Specifically, we analyzed whether proportion agriculture (i.e., inverse of “landscape quality”) was higher for these sites. In the figure below you can see that the sites that were excluded from analyses due to insufficient pollen collection were not different in terms of landscape context:

We do see a trend towards excluded colonies having a lower maximum weight (estimated mean difference: 117 g; $X^2 = 4.144$, $P = 0.0417$):

Unfortunately, we cannot explore “how might this have been related to ... pesticide loading” because we acquire pesticide information from pollen samples. Based on the second figure above and our main analyses of weight and pesticide risk, it is tempting to speculate that colonies unable to acquire sufficient pollen also experienced higher levels of pesticide risk.

We have experienced similar problems with pollen collection limitations and therefore inability to estimate pesticide risk when working with a solitary bee in California and there also suspected that high pesticide risk could be a reason (Rundlöf et al. 2022). In that same study, we also explored pesticide risk in relation to queen production of a bumble bee and relied on returning foragers’ corbicular pollen as a complement to or instead of stored pollen. We now suggest this in L338:

“The latter could potentially be avoided in any future studies by complementing with concurrent collection of returning foragers’ corbicular pollen (Rundlöf et al. 2022, Knapp et al. 2023).”

Rundlöf, M., Stuligross, C., Lindh, A., Malfi, R. L., Burns, K., Mola, J. M., ... & Williams, N. M. (2022). Flower plantings support wild bee reproduction and may also mitigate pesticide exposure effects. *Journal of Applied Ecology*, 59(8), 2117-2127.

Knapp, J. L., Nicholson, C. C., Jonsson, O., de Miranda, J. R., & Rundlöf, M. (2023). Ecological traits interact with landscape context to determine bees’ pesticide risk. *Nature Ecology & Evolution*, 7(4), 547-556.

2. L323-327: The authors indicate that new reproductive (gynes and new males) could have left the sentinel colonies during their time in the field. This seems surprising for gynes as I thought it was standard practice for commercial *B. terrestris* colony boxes to come fitted with a queen excluder to prevent new gynes from being able to exit the colonies. If this is not/ no longer the case, it would have been simple enough to restrict the nest entrance diameter of all colonies in the field to prevent gynes leaving the colonies – thereby giving a better indication of new queen (gyne) production on a colony/ site basis. The number and sizes of gynes produced are highly likely to be one of the significant population bottlenecks for bumblebees in their annual life-cycle.

Good point. None of our colonies had queen excluders on them. The reason for not using queen excluders was that colonies were weighed/inspected infrequently, so if they had queen excluders they may be totally drained of resource stores due to lots of trapped queens with no possibility to escape and forage for themselves. Unfortunately, we cannot go back and restrict nest entrances, but we will keep this in mind for future work where the scale of the work allows for more frequent inspections.

3. L376-377: “We used the LD50s for adult *A. mellifera* because there is incomplete toxicity data for other bee species, and where there are data, inter-taxa correlation is high^{18,19}.” The lack of lethal endpoint toxicity information for non-*Apis* bee species is clearly an issue to be addressed in our field of research – it would be much better to have been able to use contact and oral LD50 measurements for *Bombus terrestris* in these pesticide risk calculations. I am a little concerned that there isn’t a wider discussion (either here or in the supplemental information) about the potential variability in toxicity for the same pesticides across different bee taxa, and the potential limitations of only using toxicity data for *Apis mellifera* for such risk calculations. For example, Arena & Sgolastra’s (2014) meta-analysis showed a range of sensitivity in bee taxa (R from 0.001 to 2085.7) suggesting that in some cases honeybees are 1000x more sensitive to a pesticide than the other bee species, or in other cases that the other bee species was more than 2000x time more sensitive than a honey bee. Clearly the relative toxicity varies among bee taxa and active ingredients/ pesticide groups.

*We agree that *B. terrestris* LD_{50s} would have been fantastic to use. However, it is important to remember that we use toxicity data (in this case, from the more data-rich *A. mellifera*) to create a metric of pesticide-related risk relative to the colonies used across our study. Thus, these risk values are not absolute. Considering trends in bee species' toxicity concerning body size (EFSA 2023), risk estimates for our large-bodied *B. terrestris*' are precautionary at the bee individual level. Thus, our risk estimates should represent smaller-bodied species - the majority of bee taxa (L136)*

European Food Safety Authority (EFSA), Adriaanse, P., Arce, A., Focks, A., Ingels, B., Jölli, D., ... & Auteri, D. (2023). Revised guidance on the risk assessment of plant protection products on bees (*Apis mellifera*, *Bombus* spp. and solitary bees). *EFSA Journal*, 21(5), e07989.

4. Figure 1c: comparing the fitted lines in this figure with the results, I believe that the green fitted line represents a significant negative relationship (for apple crops) – however, the yellow fitted line (for OSR) is basically a flat line. Is the latter a statistically significant relationship? If not, this needs to be more explicitly specified in the legend (and possibly indicated graphically with a dashed fitted line rather than a solid one).

Your interpretation is correct, and this is another good point. We now report the slope estimate and confidence intervals for the significant negative trend in apple (L85):

“Colony weight change was smaller with increasing pesticide risk where apple was the focal crop (slope estimate [95% CI]: -0.13 [-0.19, -0.07]) but not at the more resource-yielding oilseed rape²¹ (0.02 [-0.06, 0.08]; Fig. 1c), suggesting...”

We do not use a dashed line in Figure 1c to indicate whether slopes are different than 0 to avoid confusion with dashed lines in the other figure panels.

Reviewer Reports on the First Revision:

Referees' comments:

Referee #1 (Remarks to the Author):

The authors have produced a very full response to reviewer comments including new analyses and figures. The manuscript is much improved and I like the new SPG analysis.

The authors have certainly responded fully to the points I made in my first review. It also looks like the responses to other reviewers are adequate – but those initial reviewers are better placed to comment on this.

Two remaining comments from me:

1) The term “weight change” is confusing as an endpoint because “change” could mean increase or decrease. I think it would be much better to use the phrase “weight gain” throughout. The word “change” is ambiguous because it could mean positive or negative whereas “gain” is clearly an increase in weight. Some colonies might have lost weight but that is OK because the gain would be negative. Particularly in the section from L76-L80. EXAMPLE 1: it says “Change in colony weight – a metric inclusive of bees, brood, and food – also decreased with increasing pesticide risk,..” but would be clearer to say “Colony weight gain – a metric inclusive of bees, brood and food – also decreased.....” EXAMPLE 2: “Colony weight change was smaller with increasing pesticide risk ...” would be better as “Colony weight gain was smaller with increasing pesticide risk”. And so on throughout the text, and the figures and tables.

2) Figure S5 – you say the symbols are circles and triangles. But I see circles and diamonds?

Referee #2 (Remarks to the Author):

I have now gone through my original comments and your very comprehensive responses. I am satisfied that you have either addressed the issues that I raised (particular happy with you looking at the <25 and >90% percentile responses and including queen production) or have provided a justifiable rebuttal to me questions. Given this and in view of the extensive questions I originally raised that you have dealt with I am now satisfied that this paper is suitable for publication in Nature. I believe this to be an important study of international relevance. A great job. Ben Woodcock

Referee #3 (Remarks to the Author):

The authors have done an excellent job of revising this manuscript in response to mine and the other reviewers' comments. I appreciate the care with which they explained how and why they've made the changes that they have (in their extensive rebuttal letter). As a result, I think the manuscript is substantially strengthened. This work provides a clear novel perspective to our understanding of how field exposure to pesticide mixtures could affect bumblebees in a real-world context – something that is both highly novel and potentially impactful for our understanding of the risks posed to essential insect pollinators (and also other non-target beneficial insects).

I have some queries and concerns about the study and the manuscript that I would like the authors to consider going forward:

Fig 1b: what is indicated by the left hand y-axis marks for this figure panel? If these do not

indicate anything, then please remove them.

L73-76: "We found that increasing pesticide risk reduced bumble bee colony production (summed enclosed and closed cocoons of all castes; see Methods), and this effect was moderated by an interaction with the proportion of cropland in the surrounding landscape (Fig. 1b; Table 1; GLMM: $\chi^2(1, 307) = 5.46, P = 0.019$)." I am a little confused by the use of the wording "this effect was moderated by an interaction with the proportion of cropland in the surrounding landscape." Do you mean that colonies facing a given pesticide risk produced more or fewer bees (i.e. increased or decreased colony production) in response to more cropland within their foraging landscape? My best interpretation of this as written is that more cropland in the landscape reduces the severity of the impact of pesticide risk, but this doesn't really agree with lines 83-85 (suggesting my interpretation here is not what you intended). Could you please revise the wording here to make your meaning less potentially ambiguous?

L79: Following on from the previous comment, similar language is used here "...surrounding landscape (Fig1d; LMM: $\chi^2(1, 307) = 10.60, P = 0.001$) moderated this effect (Table 1...." Please consider revising this wording to enhance clarity/ reduce potential ambiguity.

L148-150: "Our results show that ambitious sustainability goals related to pesticide reduction – objectives of the COP 15 meeting on the Convention on Biological Diversity⁴⁴ and the European Farm to Fork strategy⁴⁵ – would benefit bee populations, and potentially the pollination services they provide." It might be appropriate to cite the Stanley et al. (2015) study that supports the view that field-relevant pesticide exposure can affect the pollination services provided by *Bombus terrestris* to apple crops.

Stanley, D. A., et al. Neonicotinoid pesticide exposure impairs crop pollination services provided by bumblebees. *Nature* 528, 548-550 (2015).

L184: Article number is missing here. Please replace "1-13" with it here.

L189: Insert volume and page numbers here (should be "7, 547–556").

L191: replace "00, 1–11" with appropriate volume and page numbers.

L197: replace "1-7" with the article number ("51") here.

L206: Replace "1-8" with the article number ("12459") here.

L210: add article number here – should be "e06607".

L212: Please include appropriate volume and page numbers here.

L215: Article number is missing.

L218: Article number is missing.

L220: Article number is missing.

L224: Replace "1-6" with article number please.

L233: Article number is missing.

L235: Volume number is missing here.

L237: Article number is missing.

L239: "Apis mellifera" should be in italics.

L240: Replace "1-21" with article number please.

L242: "Bombus" should be in italics.

L243: Article number is missing.

L245: update volume and page number format and delete "Preprint at <https://doi.org/10.1371/journal.pone.0096580>".

L260: "Bombus terrestris" should be in italics.

L263: Replace "1-5" with article number please.

L264: Update formatting of reference information to correct style here.

L267: Update formatting of reference information to correct style here.

L270: Article number is missing.

L355: Do you definitely mean "thermophilous", rather than say "entomophilous", here?

L434: Journal title and volume number missing. Should be "Sustainability 14, 10549".

L438: Article number is missing.

L447: Volume and page/article numbers missing.

L457: page/article numbers missing.

L459: Please provide appropriate source information for this reference.

L461: replace "00, 1-11" with appropriate volume and page numbers.

L463: Insert volume and page numbers here (should be "7, 547-556").

L484: "R J" seems like it might not be the full journal/ source title.

L486-7: Can you provide page/article numbers for this reference?

L490: replace "1-8" with the actual page/ article numbers?

Table S2: check the percentage figures in the fourth column "Quantification frequency" as some of these appear to be incorrect. For example, line 1 (Indoxacarb) has 17 detections from 106 samples, which is 16% (not 17%) of detections. Similarly, for line 3 (Chlorpyrifos) there are 9 detections (from 106 samples) that equates to 8% (rather than 9%) of detections. Please review this table carefully.

L565: I suggest changing "Proportion" to "Percentage" in the first sentence of the figure legend.

L607-608: "Using a suggested SPG for bumble bees of 10% reduction in colony weight⁴² (yellow line), 60% of the remaining colonies in our study exceed this." I agree that 60% of colonies

exceed this threshold of 10% reduction in colony weight, but it might also be helpful to mention that means these colonies fall below the yellow horizontal line in panel a.

Author Rebuttals to First Revision:

Referees' comments:

Referee #1 (Remarks to the Author):

The authors have produced a very full response to reviewer comments including new analyses and figures. The manuscript is much improved and I like the new SPG analysis.

The authors have certainly responded fully to the points I made in my first review. It also looks like the responses to other reviewers are adequate – but those initial reviewers are better placed to comment on this.

We thank this reviewer for their time in thoroughly reviewing our response.

Two remaining comments from me:

1) The term “weight change” is confusing as an endpoint because “change” could mean increase or decrease. I think it would be much better to use the phrase “weight gain” throughout. The word “change” is ambiguous because it could mean positive or negative whereas “gain” is clearly an increase in weight. Some colonies might have lost weight but that is OK because the gain would be negative. Particularly in the section from L76-L80. EXAMPLE 1: it says “Change in colony weight – a metric inclusive of bees, brood, and food – also decreased with increasing pesticide risk,…” but would be clearer to say “Colony weight gain – a metric inclusive of bees, brood and food – also decreased……” EXAMPLE 2: “Colony weight change was smaller with increasing pesticide risk …” would be better as “Colony weight gain was smaller with increasing pesticide risk”. And so on throughout the text, and the figures and tables.

Good suggestion, ‘weight gain’ is more understandable. We have revised this term throughout the manuscript, including updating figure axes labels and tables.

2) Figure S5 – you say the symbols are circles and triangles. But I see circles and diamonds?

Good catch. Change made

Referee #2 (Remarks to the Author):

I have now gone through my original comments and your very comprehensive responses. I am satisfied that you have either addressed the issues that I raised (particular happy with you looking at the <25 and >90% percentile responses and including queen production) or have provided a justifiable rebuttal to me questions. Given this and in view of the extensive questions I originally

raised that you have dealt with I am now satisfied that this paper is suitable for publication in Nature. I believe this to be an important study of international relevance. A great job. Ben Woodcock
These comments were very helpful. We are thankful for the reviewer pushing us to clarify our thinking and terminology around risk. We also find the revised SPG analyses to be much improved.

Referee #3 (Remarks to the Author):

The authors have done an excellent job of revising this manuscript in response to mine and the other reviewers' comments. I appreciate the care with which they explained how and why they've made the changes that they have (in their extensive rebuttal letter). As a result, I think the manuscript is substantially strengthened. This work provides a clear novel perspective to our understanding of how field exposure to pesticide mixtures could affect bumblebees in a real-world context – something that is both highly novel and potentially impactful for our understanding of the risks posed to essential insect pollinators (and also other non-target beneficial insects).

We thank this reviewer for their excellent perspectives on our work. We address their remaining concerns below.

I have some queries and concerns about the study and the manuscript that I would like the authors to consider going forward:

Fig 1b: what is indicated by the left hand y-axis marks for this figure panel? If these do not indicate anything, then please remove them.

These axis ticks are redundant of the right hand y-axis ticks. We have removed them

L73-76: “We found that increasing pesticide risk reduced bumble bee colony production (summed enclosed and closed cocoons of all castes; see Methods), and this effect was moderated by an interaction with the proportion of cropland in the surrounding landscape (Fig. 1b; Table 1; GLMM: $\chi^2(1, 307) = 5.46, P = 0.019$.” I am a little confused by the use of the wording “this effect was moderated by an interaction with the proportion of cropland in the surrounding landscape.” Do you mean that colonies facing a given pesticide risk produced more or fewer bees (i.e. increased or decreased colony production) in response to more cropland within their foraging landscape? My best interpretation of this as written is that more cropland in the landscape reduces the severity of the impact of pesticide risk, but this doesn't really agree with lines 83-85 (suggesting my interpretation

here is not what you intended). Could you please revise the wording here to make your meaning less potentially ambiguous?

Yes, we can see the confusion. We do not mean 'moderate' as in 'to make less intense or extreme'. What we mean is that this effect depends on the proportion of cropland. To avoid confusion we have revised our wording and instead say "modified".

L79: Following on from the previous comment, similar language is used here "...surrounding landscape (Fig1d; LMM: $\chi^2(1, 307) = 10.60, P = 0.001$) moderated this effect (Table 1...." Please consider revising this wording to enhance clarity/ reduce potential ambiguity.

See above

L148-150: "Our results show that ambitious sustainability goals related to pesticide reduction – objectives of the COP 15 meeting on the Convention on Biological Diversity⁴⁴ and the European Farm to Fork strategy⁴⁵ – would benefit bee populations, and potentially the pollination services they provide." It might be appropriate to cite the Stanley et al. (2015) study that supports the view that field-relevant pesticide exposure can affect the pollination services provided by *Bombus terrestris* to apple crops.

Yes. Good addition. We have added this reference Stanley, D. A., et al. Neonicotinoid pesticide exposure impairs crop pollination services provided by bumblebees. Nature 528, 548-550 (2015).

For the comments regarding the formatting of works cited (L184-270 & L434-490) we have updated our references. These should all be correct now, but are happy to work with copyeditors to make sure

L184: Article number is missing here. Please replace "1-13" with it here.

done

L189: Insert volume and page numbers here (should be "7, 547–556").

done

L191: replace "00, 1–11" with appropriate volume and page numbers.

done

L197: replace "1-7" with the article number ("51") here.

done

L206: Replace “1-8” with the article number (“12459”) here.

done

L210: add article number here – should be “e06607”.

done

L212: Please include appropriate volume and page numbers here.

done

L215: Article number is missing.

done

L218: Article number is missing.

done

L220: Article number is missing.

done

L224: Replace “1-6” with article number please.

done

L233: Article number is missing.

done

L235: Volume number is missing here.

done

L237: Article number is missing.

done

L239: “*Apis mellifera*” should be in italics.

done

L240: Replace “1-21” with article number please.

done

L242: “*Bombus*” should be in italics.

done

L243: Article number is missing.

done

L245: update volume and page number format and delete “Preprint at

<https://doi.org/10.1371/journal.pone.0096580>”

done

L260: “*Bombus terrestris*” should be in italics.

done

L263: Replace “1-5” with article number please.

done

L264: Update formatting of reference information to correct style here.

done

L267: Update formatting of reference information to correct style here.

done

L270: Article number is missing.

done

L355: Do you definitely mean “thermophilous”, rather than say “entomophilous”, here? *T*

his term comes from palynology and it refers to species existing in warmer environments than those otherwise indicated by the pollen record

Allaby, M. (Ed.). (2012). A dictionary of plant sciences. Oxford University Press, USA.

L434: Journal title and volume number missing. Should be “Sustainability 14, 10549”.

done

L438: Article number is missing.

done

L447: Volume and page/article numbers missing.

done

L457: page/article numbers missing.

done

L459: Please provide appropriate source information for this reference.

done

L461: replace “00, 1–11” with appropriate volume and page numbers.

done

L463: Insert volume and page numbers here (should be “7, 547–556”).

done

L484: "R J" seems like it might not be the full journal/ source title.

done

L486-7: Can you provide page/article numbers for this reference?

Yes. done

L490: replace "1-8" with the actual page/ article numbers?

done

Table S2: check the percentage figures in the fourth column "Quantification frequency" as some of these appear to be incorrect. For example, line 1 (Indoxacarb) has 17 detections from 106 samples, which is 16% (not 17%) of detections. Similarly, for line 3 (Chlorpyrifos) there are 9 detections (from 106 samples) that equates to 8% (rather than 9%) of detections. Please review this table carefully.

Good catch. We have updated values in Table 2 and now Supplementary Table 2 so that they are reporting detection frequencies from all samples (106), rather than only those samples in which compounds were detected (103).

L565: I suggest changing "Proportion" to "Percentage" in the first sentence of the figure legend.

Indeed. Good catch

L607-608: "Using a suggested SPG for bumble bees of 10% reduction in colony weight⁴² (yellow line), 60% of the remaining colonies in our study exceed this." I agree that 60% of colonies exceed this threshold of 10% reduction in colony weight, but it might also be helpful to mention that means these colonies fall below the yellow horizontal line in panel a.

We believe that the text in yellow within the figure panel ("n = 143 (60%)") makes this visually clear